# Zona incerta distributes a broad movement signal that modulates behavior

**Sebastian Hormigo, Ji Zhou, Dorian Chabbert, Sarmad Sajid, Natan Busel, Manuel Castro-Alamancos***

Department of Neuroscience, University of Connecticut School of Medicine, Farmington, United States

**Abstract** The zona incerta is a subthalamic nucleus made up mostly of GABAergic neurons. It has wide-ranging inputs and outputs and is believed to have many integrative functions that link sensory stimuli with motor responses to guide behavior. However, its role is not well established perhaps because few studies have measured the activity of zona incerta neurons in behaving animals under different conditions. To record the activity of zona incerta neurons during exploratory and cue-driven goal-directed behaviors, we used electrophysiology in head-fixed mice moving on a spherical treadmill and fiber photometry in freely moving mice. We found two groups of neurons based on their sensitivity to movement, with a minority of neurons responding to whisker stimuli. Furthermore, zona incerta GABAergic neurons robustly code the occurrence of exploratory and goal-directed movements, but not their direction. To understand the function of these activations, we performed genetically targeted lesions and optogenetic manipulations of zona incerta GABAergic neurons during exploratory and goal-directed behaviors. The results showed that the zona incerta has a role in modulating the movement associated with these behaviors, but this has little impact on overall performance. Zona incerta neurons distribute a broad corollary signal of movement occurrence to their diverse projection sites, which regulates behavior.

***For correspondence:**
mcastro@uchc.edu

## eLife assessment

This **important** study uses a range of technical approaches to investigate the responses of zona incerta neurons to movement and sensory stimuli. The majority of neurons exhibited movement related activity but only a small proportion were modulated by whisker deflections. The major conclusion of the study is that the zona incerta distributes a general motor signal. The evidence supporting this claim is **solid**, although the study would be improved by greater transparency and discussion of experimental methods and histological verification of recording sites, viral spread, and which territories of the zona incerta were investigated. The work will be of interest to behavioral and physiological neuroscientists.

## Introduction

The zona incerta is a GABAergic nucleus in the subthalamus with wide-ranging inputs and outputs (*Mitrofanis, 2005*), but its precise role is still unclear. The zona incerta has been associated with a variety of functions, including regulating thalamocortical and thalamostriatal transmission through its GABAergic projections to the posterior thalamus (*Trageser and Keller, 2004*; *Lavallée et al., 2005*; *Trageser et al., 2006*; *Urbain and Deschênes, 2007*; *Watson et al., 2015*), contributing to binge eating through its projections to the paraventricular thalamus (*Zhang and van den Pol, 2017*), and

promoting feeding through projections to the ventral tegmental area (*de Git et al., 2021*). The zona incerta has also been implicated in defensive behaviors through its projections to the periaqueductal gray (*Chou et al., 2018*), regulating cross-modality enhancement of flight through its projections to the posterior medial nucleus of thalamus (*Wang et al., 2019*), and promoting sleep through subsets of its GABAergic neurons (*Liu et al., 2017*). In addition, the zona incerta has been linked to other behaviors, such as mediating Pavlovian fear conditioning and anxiety (*Li et al., 2021*; *Zhou et al., 2018*; *Venkataraman et al., 2019*), controlling predatory hunting (*Zhao et al., 2019*), novelty seeking (*Ogasawara et al., 2022*), itch (*Li et al., 2022*), and neuropathic pain (*Masri et al., 2009*; *Hu et al., 2019*; *Singh et al., 2022*). It has also been suggested as a site to alleviate essential tremor with deep brain stimulation (*Plaha et al., 2006*) and implicated in motor control (*Mitrofanis, 2005*). Furthermore, projections from the zona incerta to the midbrain tegmentum can control learned, cued avoidance locomotor movement, but this function is not essential for driving the behavior (*Hormigo et al., 2019*; *Hormigo et al., 2020*). Zona incerta has been proposed to have broad integrative functions (*Mitrofanis, 2005*; *Wang et al., 2020a*), including serving as an inhibitory switchboard for behavioral control (*Fratzl and Hofer, 2022*).

Given the wide range of functions attributed to the zona incerta, it is possible that it performs a basic function that applies to many behaviors, such as monitoring ongoing movement. In the brain, internal copies of motor commands, called corollary discharges or efference copy signals, can affect other brain processes, such as sensory processing (*Grüsser, 1995*; *Sommer and Wurtz, 2008*). These signals have a critical regulatory role in brain function but are typically not required to generate the behaviors they monitor or signal. For instance, when animals move, the neural activity that drives the movements (efference) must be disentangled from the neural activity driven by contact with objects (exafference) and fed back by the movement (reafference). Many brain regions can use an efference copy of ongoing movement for various control and modulatory purposes related to their functions.

To test the possibility that the zona incerta distributes information about ongoing movement to other brain areas that employ it to regulate behavior, we recorded the activity of zona incerta neurons in behaving mice using cell type-specific calcium imaging, fiber photometry, and single-unit recordings in freely moving and head-fixed mice, respectively. We also employed optogenetics and genetically targeted lesions to assess the functional role of these neurons. Our findings demonstrate that zona incerta neurons integrate a broad movement signal that modulates behavior but is not required to drive it.

## Results

### Zona incerta units activate during movement

To determine the activity of individual zona incerta neurons in awake behaving mice, we used high-impedance electrodes to record from well-isolated zona incerta neurons in head-fixed mice moving on a spherical treadmill (*Figure 1A*), as previously described (*Hormigo et al., 2021c*). Using the recording electrode, a marking lesion was made at a defined depth above the zona incerta (*Figure 1B*). The brain was then extracted, sectioned, and sections around the lesion were aligned with the Allen Brain Atlas. Recorded units were assigned to the zona incerta if they were recorded at a depth below this reference point corresponding to the zona incerta in the histological tissue. *Figure 1B* shows an electrode track toward zona incerta and a marking lesion in the reference point above the zona incerta in a sagittal section (dark field) aligned with the atlas. During recordings, mice move forward in place on the treadmill spontaneously or motivated by repetitive air-puffs applied to their lower back. We also tested the responses of units to sensory stimuli, including auditory tones and whisker deflections.

As mice moved on the treadmill, we recorded units in the zona incerta area ($n = 341$). Using this dataset, we detected movement (treadmill turns), and created PSTHs around these movement peaks. We then used PCA to classify the units based on the PSTHs around movement (*Figure 1C, D*). This revealed two major groups of neurons (Class1 and Class2) that had different coding of movement and could each be further subclassified into two subgroups (a and b). Class1 neurons activated robustly during movement (*Figure 1C, D*) and were differentiated based on their firing rate; Class1a ($n = 23$, 6.7%) neurons had higher firing rates than Class1b ($n = 82$, 24.1%) neurons (Tukey $q = 12.36$, $p < 0.0001$). Class2 neurons consisted of two distinct subclasses. Class2a were somewhat inhibited during movement ($n = 80$, 23.5%) and Class2b were weakly activated during movement ($n = 156$, 45.7%). In

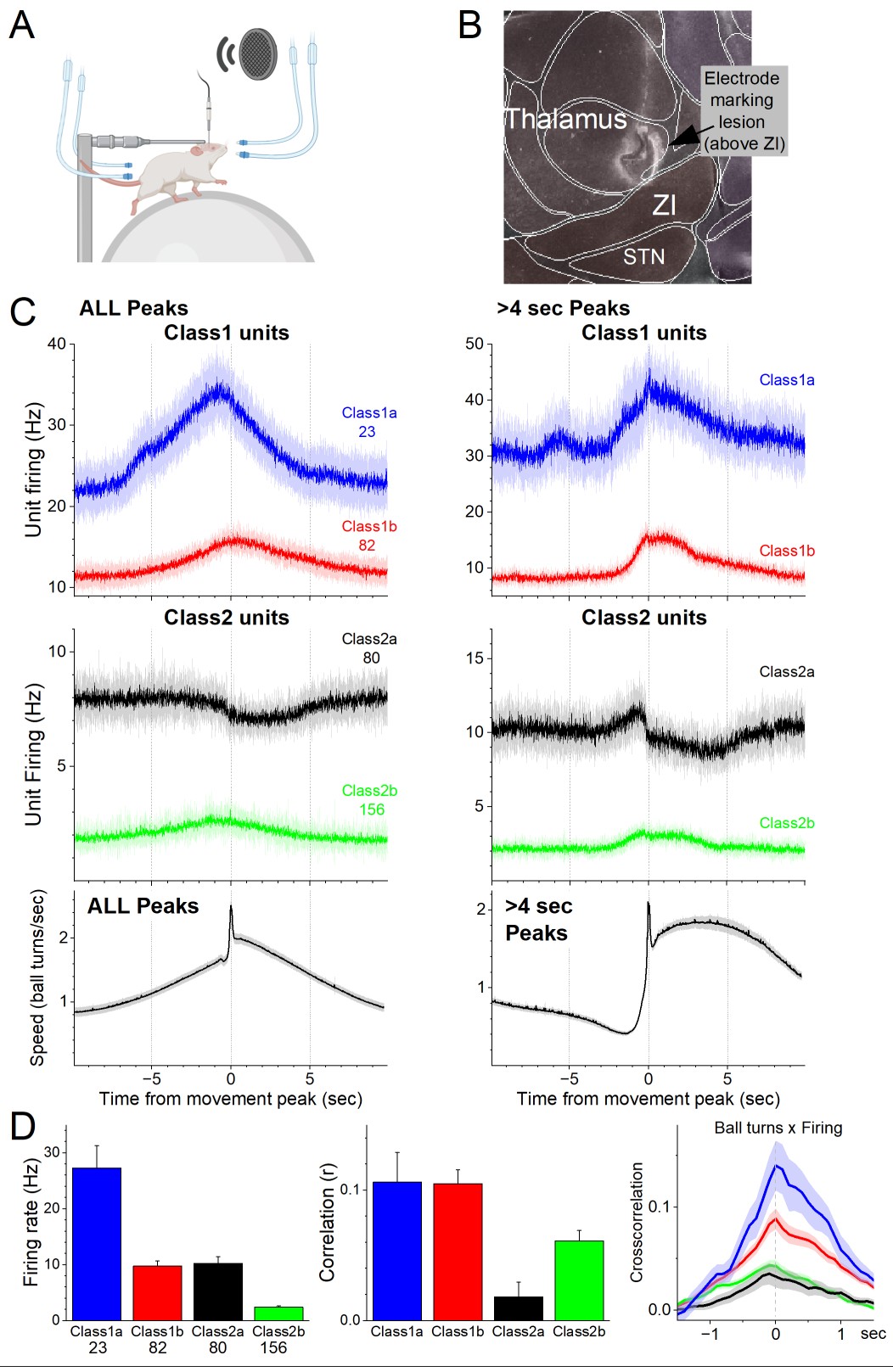

**Figure 1.** Zona incerta unit activity measured with electrophysiology in head-fixed mice on a spherical treadmill reveals two groups of neurons based on their activation during movement. (**A**) Schematic of the head-fixed spherical treadmill setup. The frontal air tubes deliver air-puffs to the whiskers, and the posterior air tubes deliver air-puffs to the lower back. (**B**) Dark-field sagittal section aligned with the Allen Brain Atlas showing a marking

*Figure 1 continued on next page*

*Figure 1 continued*

lesion to reference to point above the zona incerta below where recordings were obtained. (**C**) PSTH of unit firing (Hz) and movement (speed) traces for units classified using PCA as Class1 (upper panel) and Class2 (middle panel). Class1 units activate during movement and can be further subclassified into Class1a (blue) and Class2b (red) based on baseline firing rate. The lower panel shows the speed of the mice on the spherical treadmill (in the forward direction) for all units. Time zero corresponds to the peak of detected spontaneous movements. The left panels show all detected movement peaks, and the right panels show those peaks devoid of other peaks >4 s prior, which selects for movement onsets. (**D**) Population measures of unit firing rate (left), linear fit (correlation) between firing rate and movement (middle), and cross-correlation between firing rate and movement (right) for the different groups.

addition, Class2a neurons had higher firing rates than Class2b neurons (Tukey $q$ = 9.48, p<0.0001) but equivalent to Class1b neurons (Tukey $q$ = 0.54, p = 0.9814). Moreover, Class1a and Class1b neurons had equivalent positive correlations between speed and firing (Tukey $q$ = 0.07, p = 1). Because Class2a neurons could suppress their firing during movement they had weaker positive correlations than all the other subgroups, including Class2b neurons (Tukey $q$ = 4.42, p = 0.01), which correlated poorly with movement compared to Class1 neurons. Class1b neurons had higher correlations than Class2b neurons (Tukey $q$ = 4.54, p = 0.0079). There was no difference in recording depth between the four classes of neurons (analysis of variance [ANOVA] $F_{(3,337)}$ = 1.06, p = 0.3676).

We then explored the effects of auditory and tactile stimuli on a group of these units. Auditory stimuli consisted of a tone (8 kHz) or white noise. Tactile stimuli consisted of air-puffs applied from the front, aimed at the contralateral or ipsilateral whiskers, and from the back, aimed at the contralateral or ipsilateral lower back. The tested units ($n$ = 170) did not respond to the auditory tone, white noise, or to single air-puffs applied to the lower back (*Figure 2A*). However, a portion of the neurons ($n$ = 33, 19.4%) responded to air-puffs aimed at the contralateral whiskers (*Figure 2A*). The units with the strongest contralateral whisker-evoked responses also responded to the ipsilateral whiskers with weaker responses (*Figure 2A*). Whisker-sensitive neurons did not fall into a specific class of movement-sensitive neurons (*Figure 2B*). About half of these units belonged to Class1 ($n$ = 15/33) indicating that they activated during movement (*Figure 2B*, black). The other whisker-sensitive units ($n$ = 18/33) were poorly modulated by movement (*Figure 2B*, red). *Figure 2C* shows population measures of whisker-evoked and auditory responses for whisker-sensitive and -insensitive neurons. The effect of the sensory stimuli depended on the whisker sensitivity of the cells (mixed ANOVA interaction of Stimulus × Whisker sensitivity $F_{(3,555)}$ = 156.81, p < 0.0001). Thus, only the whisker-sensitive cells showed a stronger response to the contralateral (Tukey's $t_{(555)}$ = 24, p < 0.0001 vs the other stimuli including ipsilateral whiskers) or ipsilateral (Tukey's $t_{(555)}$ = 5, p < 0.009 vs the auditory stimuli) whisker stimulation compared to the other stimulation. For whisker-sensitive cells (*Figure 2D*), there was no interaction between whisker stimulation side and class (mixed ANOVA interaction $F_{(1,43)}$ = 1.92, p = 0.17) indicating that the responses evoked by contralateral and ipsilateral stimulation did not differ between Class1 and Class2 neurons. Although within each class, contralateral whisker stimulation produced stronger responses than ipsilateral stimulation (Tukey $t_{(43)}$ = 8.0, p < 0.0001 for Class1 and Tukey $t_{(43)}$ = 5.6, p = 0.001 for Class2). Finally, for whisker-sensitive neurons, spontaneous firing rate differed per class (*Figure 2E*; two-sample $t$-test $t_{(43)}$ = 3.7, p = 0.0006).

The electrophysiology results show that a significant portion of neurons in the zona incerta area are modulated by movement. Also, most of these neurons are poorly responsive to sensory stimulation other than very salient whisker puffs, which trigger both passive (deflection) and active (reactive) movement of the whiskers in awake mice. Thus, zona incerta activation is closely associated with movement.

## Zona incerta GABAergic neurons code movement but not its direction

Single-unit recordings in the area of the zona incerta reveal populations of neurons that activate during movement. However, these recordings do not target GABAergic neurons and may sample some neurons in the tissue surrounding the zona incerta. Therefore, we used calcium imaging fiber photometry to target GABAergic neurons in the zona incerta. To measure the activity of zona incerta GABAergic neurons, we expressed GCaMP7f (*Chen et al., 2013*) in these neurons by injecting a Cre-AAV in the zona incerta of Vgat-cre mice ($n$ = 8). Thereafter, a single optical fiber was implanted

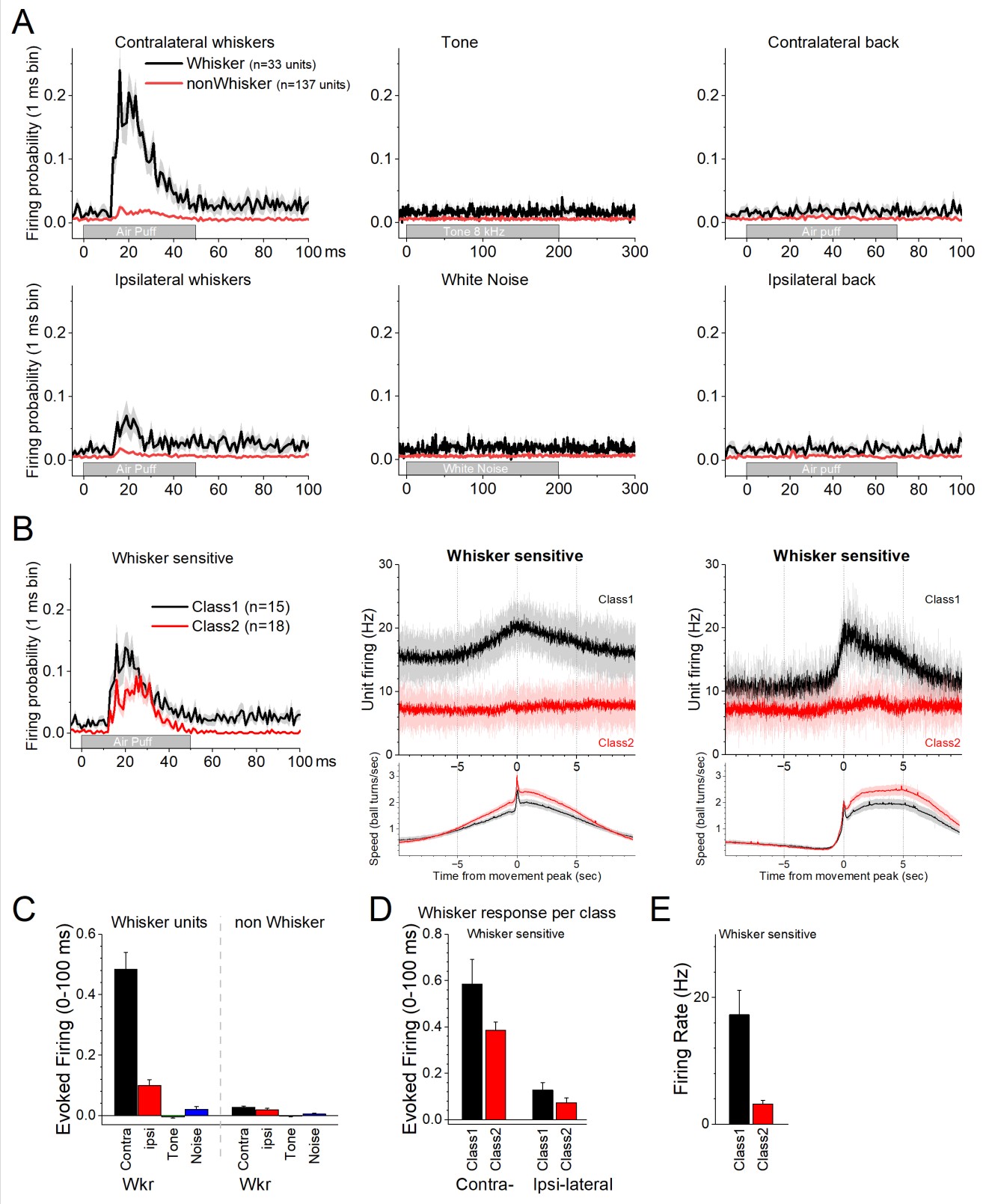

**Figure 2.** Response of zona incerta units to sensory stimuli applied to head-fixed mice on the spherical treadmill. (**A**) PSTH of unit firing evoked by air-puffs to the whiskers (left panels) or lower back (right panels), and auditory stimuli consisting of a tone (8 kHz) or white noise (middle panels). Air-puffs are delivered contralateral (top panels) or ipsilateral (bottom panels) to the recording site. The black traces correspond to whisker-sensitive units (*n* = 33). The red traces correspond to whisker non-sensitive units (*n* = 137). (**B**) The whisker-sensitive cells in A are separated according to their class as per

*Figure 2 continued on next page*

Figure 2 continued

**Figure 1.** Class1 units are sensitive to movement on the treadmill ($n$ = 15), while Class2 units are not ($n$ = 18). Thus, whisker-sensitive neurons fall under both classes of neurons. The right panels show the PSTH of unit firing and movement for the units on the left panels. (**C**) Population measures of evoked firing for the units and stimuli in A. (**D**) Population measures of evoked firing for the units and whisker stimuli in B. (**E**) Population measures of firing rate per class for the whisker-sensitive units.

at the injection site to monitor neural activity by imaging calcium signals with fiber photometry, as previously described (*Hormigo et al., 2021a*; *Zhou et al., 2023*). *Figure 3A* shows optical fiber tracks in the zona incerta of two mice targeting GCaMP-expressing GABAergic neurons; it is estimated that the optical fiber images a volume ($2.5 \times 10^7$ mm$^3$) that extends ~200 mm from its ending (*Pisanello et al., 2019*).

In freely moving mice, we continuously measured calcium fluorescence ($F/F_o$) and spontaneous movement while mice explored an arena. Movement measures were derived from head markers. To establish the relation between movement and zona incerta $F/F_o$ activation, we calculated cross-correlations between the continuous variables (*Figure 3B*, upper). Overall movement, including both rotational and translational components, were strongly correlated with zona incerta GABAergic neuron activation. We also calculated the linear fit between the movement and $F/F_o$ by integrating over a 200-ms window, which revealed a strong linear positive correlation (*Figure 3B*, lower). This relation was absent when one of the variables in the pair was shuffled (*Figure 3B*, lower).

To further evaluate the relationship between movement and zona incerta activation, we detected spontaneous movements and time extracted the continuous variables around the detected movements peaks (*Figure 3C*). The detected movements were classified in three categories. The first category includes all peaks (*Figure 3C*, black traces), which revealed a strong zona incerta GABAergic neuron activation in relation to movement. The second category includes movements that had no detected peaks 3 s prior (*Figure 3C*, red traces), which essentially extracts movement onsets from immobility. This revealed a sharp activation in association with movement onset. The third category sampled the peaks by averaging every >5 s to eliminate from the average the effect of closely occurring peaks (*Figure 3C*, cyan traces). This category includes movement increases from ongoing baseline movement (instead of movement onsets from immobility) and showed a strong activation of zona incerta neurons. For the three categories of movement peaks, the zona incerta activation around movement was significant compared to baseline activity (Tukey p < 0.0001). Thus, zona incerta neurons discharge in relation to the occurrence of movement.

We next determined if the zona incerta activation during movement depends on the direction of the head movement in the ipsiversive or contraversive direction. *Figure 4A* shows head movement peaks in the contraversive (*Figure 4A*, cyan) and ipsiversive (*Figure 4A*, red) directions versus the recorded zona incerta neurons. While the detected movements were opposite in direction and similar in amplitude, the zona incerta GABAergic neuron activation was strikingly similar in both directions. There was no significant difference in the amplitude, area, or timing of the $F/F_o$ activation (*Figure 4B*) indicating that zona incerta neurons do not code the orienting direction of head movements. Instead, zona incerta GABAergic neurons discharge in relation to ongoing movement in either direction providing a generalized movement signal to the distributed network of sites where they project. The activation of zona incerta GABAergic neurons in relation to movement direction contrasts with the activation of neurons in other brain areas; for example, superior colliculus neurons measured with the same methods discriminate movement direction (*Zhou et al., 2023*).

## Zona incerta GABAergic neuron activation by auditory and whisker stimuli

Next, we conducted auditory mapping sessions to test if auditory tones activated zona incerta GABAergic neurons in freely behaving mice. During auditory mapping sessions (37 sessions in 8 mice; *Figure 5A, B*), mice were placed in a small cage (half the size of a shuttle box) and 10 auditory tones of different saliencies, defined by sound pressure level (SPL in dB; low and high, ~70 and ~85 dB) and frequency (4, 6, 8, 12, and 16 kHz), were presented in random order (1 s tones every 4–5 s, each repeated 10 times). The calcium signal evoked in zona incerta neurons by the tones (0–1.5 s window) depended on SPL (TwoWayRMAnova $F_{(1,36)}$ = 92.21, p < 0.0001) and frequency ($F_{(4,144)}$ = 22.18, p < 0.0001), but not their interaction ($F_{(4,144)}$ = 1.8, p = 0.13). In general, higher SPL and frequency tones

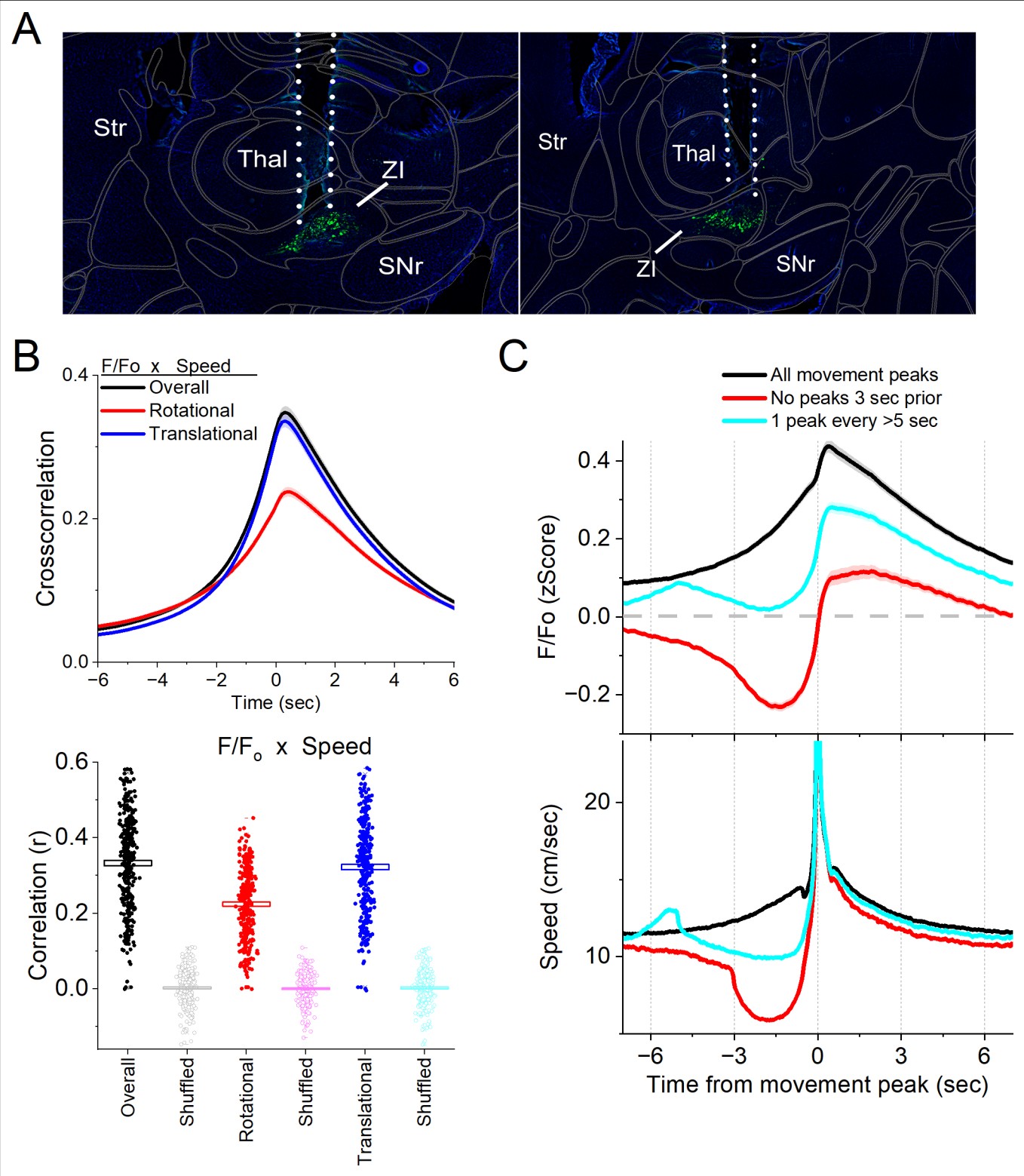

**Figure 3.** Calcium imaging fiber photometry reveals that GABAergic zona incerta neurons activate during spontaneous exploratory movement. (**A**) Parasagittal sections showing the optical fiber tract reaching zona incerta and GCaMP7f fluorescence expressed in GABAergic neurons around the fiber ending for two animals. The sections were aligned with the Allen Brain Atlas. (**B**) Cross-correlation between movement and zona incerta $F/F_o$ for the overall (black traces), rotational (red), and translational (blue) components (upper panel). Per session (dots) and mean ± standard error of the mean (SEM) (rectangle) linear fit (correlation, $r$) between overall movement and zona incerta $F/F_o$, including the rotational and translational components (lower

*Figure 3 continued on next page*

*Figure 3 continued*

panel). The lighter dots show the linear fits after scrambling one of variables (lower panel, shuffled). (**C**) $F/F_o$ calcium imaging time extracted around detected spontaneous movements. Time zero represents the peak of the movement. The upper traces show $F/F_o$ mean ± SEM of all movement peaks (black), those that had no detected peaks 3 s prior (red), and peaks taken at a fixed interval >5 s (cyan). The lower traces show the corresponding movement speed for the selected peaks.

produced stronger zona incerta activation (*Figure 5*). However, these same effects were observed on movement (overall speed). Thus, the movement evoked by the tones also depended on tone SPL ($F(1,36) = 89$, $p < 0.0001$) and frequency ($F(4,144) = 16.66$, $p < 0.0001$), but not their interaction ($F(4,144) = 0.62$, $p = 0.65$). The movement evoked by the tones consisted of both rotational and translational components (*Figure 5B*), and each of these components showed the same effects as overall movement (SPL or frequency $p < 0.0001$, interaction $p > 0.18$). The results indicate that zona incerta GABAergic neurons respond robustly to salient auditory tones in association with movement.

To decipher if the activation of zona incerta GABAergic neurons was sensory (auditory) or motor (movement caused by the auditory stimulus), we performed calcium imaging fiber photometry in the same animals that were head fixed and loosely restrained under three conditions: awake, anesthetized (isoflurane), and recovery from anesthesia. Three different stimuli were presented during these conditions: auditory tones (8 and 12 kHz), white noise, and air-puffs aimed to deflect the contralateral whiskers. *Figure 5C* shows the $F/F_o$ responses evoked during the three different conditions. Auditory tones evoked responses in awake mice that were loosely restrained but this condition was accompanied by significant movement including slight movement of the limbs, and extensive facial movements including whisker movements. Since these movements could be driving the activation of these neurons to the tones, we used light isoflurane anesthesia to abolish movements associated with the stimuli. This led to a reversible strong suppression or abolishment of zona incerta activation evoked by the tones (Tukey $t(8) = 4.3$, $p < 0.008$, Head Fixed vs +Isoflurane; 0–2 s area), white noise (Tukey $t(8) = 12.5$, $p < 0.0001$, Head Fixed vs +Isoflurane), and whisker stimuli (Tukey $t(8) = 8.07$, $p = 0.001$, Head Fixed vs +Isoflurane). Although arousal and movement were not dissected in the present experiment (this would likely require paralyzing and ventilating the animal), the results indicate that activation of zona incerta neurons by sensory stimulation is primarily associated with states when sensory-evoked movement is also present.

## Zona incerta neurons activate during goal-directed avoidance movement

Since zona incerta GABAergic neurons activate during movement, we explored their activation while performing a series of cued (signaled) avoidance tasks in which mice move to prevent an aversive unconditioned stimulus (US) (*Figure 6A*). We employed four procedures termed AA1, AA2, AA3, and AA4. AA1 is a basic signaled active avoidance procedure where mice learn to avoid an aversive US (foot-shock and white noise) by shuttling between two compartments in a cage during an avoidance interval (7 s) signaled by the auditory CS (8 kHz tone). If the animal fails to avoid during the CS, the US is presented causing a rapid escape to the other compartment, which eliminates the US and initiates a random intertrial interval until the next trial. In AA1, mice are free to shuttle between the cage compartments during the random intertrial interval that separates CS presentations; intertrial crossings (ITCs; cyan filled bars; *Figure 6B*) have no consequence (*Hormigo et al., 2021b*; *Zhou et al., 2022*). The AA2 procedure is identical to AA1 but ITCs are punished; mice must inhibit their tendency to shuttle between CS presentations during the intertrial interval (unsignaled passive avoidance). This produces a more cautious behavioral state compared to AA1 (*Zhou et al., 2022*). The AA3 procedure is a more challenging discrimination procedure with two different CS's presented randomly, and ITCs are not punished. CS1 (8 kHz) is identical to the CS in AA1/2 (signaled active avoidance), but during CS2 (4 kHz) mice must passively avoid punishment by not shuttling (signaled passive avoidance). In the AA4 procedure, CS1 (8 kHz), CS2 (10 kHz), and CS3 (12 kHz) were associated with 4-, 7-, and 15-s avoidance interval durations, respectively. In AA4, mice shift their avoidance response latencies, so that the CS that signals a shorter avoidance interval drives faster avoidance response latencies (*Hormigo et al., 2021b*). After AA4, we conducted additional sessions in which the US (foot-shock and/or white noise) was presented unsignaled; the unsignaled US drives escape responses on every trial.

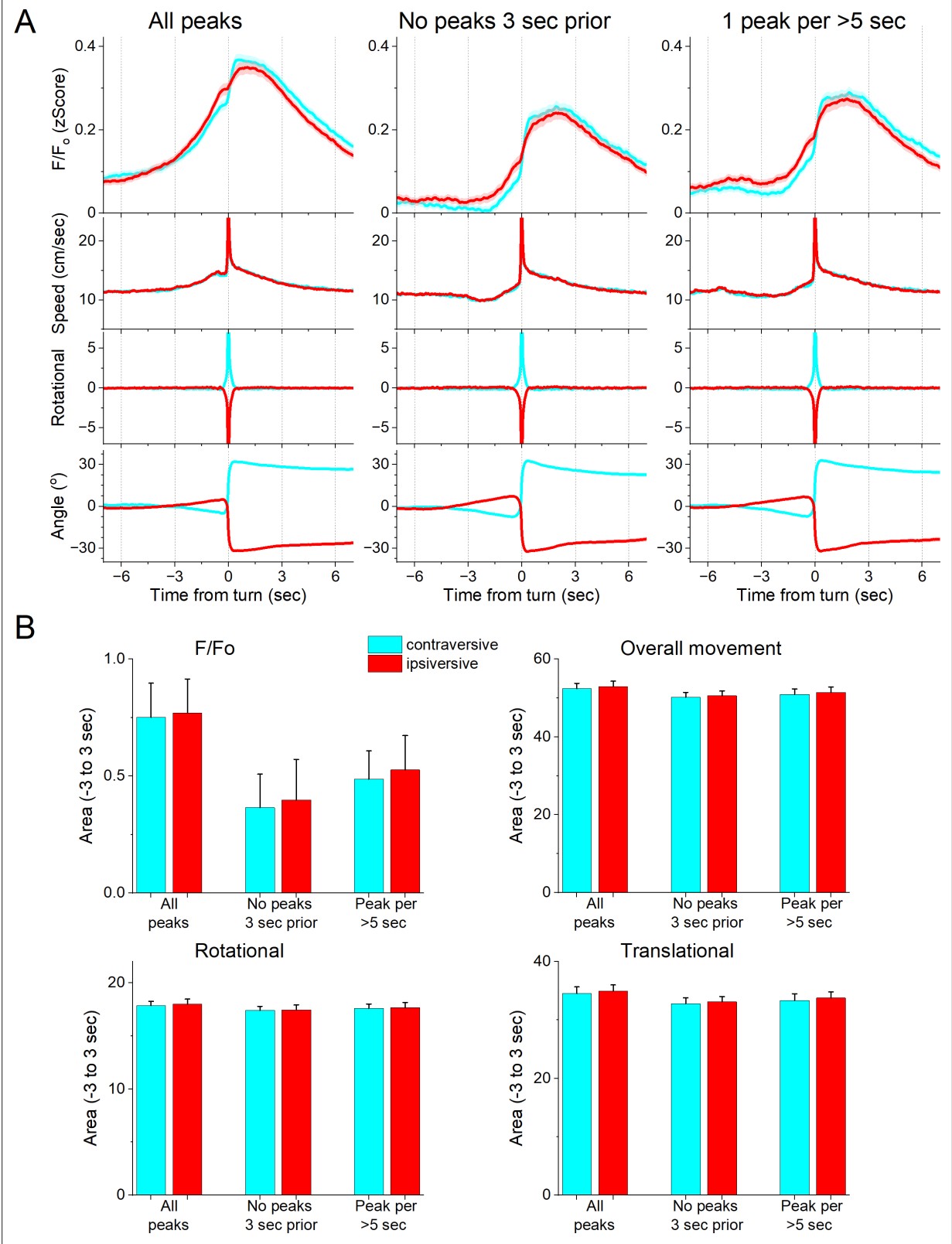

**Figure 4.** GABAergic zona incerta neurons do not code the direction of spontaneous exploratory turning movements. (**A**) $F/F_o$ calcium imaging, overall movement, rotational movement, and angle of turning direction for detected movements classified by the turning direction (ipsiversive and contraversive; red and cyan) versus the side of the recording (implanted optical fiber). At time zero, the animals spontaneously turn their head in the indicated direction. The columns show all turns (left), those that included no turn peaks 3 s prior (middle), and peaks selected at a fixed interval >5 s

*Figure 4 continued on next page*

Figure 4 continued

(right). Note that the speed of the movements was similar in both directions. (**B**) Population measures (area of traces 3 s around the detected peaks) of $F/F_o$ and movement (overall, rotational, and translational) for the different classified peaks. None of the measures were different between ipsiversive and contraversive movements.

We measured $F/F_o$ in zona incerta neurons as mice ($n$ = 8 mice) performed the different avoidance procedures. *Figure 6B* shows the behavioral performance of the animals in these tasks including the percentage of avoids (open black circles), the avoid latencies (closed orange circles), and the number of ITCs (cyan bars). As previously shown, animals perform a large percentage of active avoids during the AA1, AA2, AA3-CS1, and AA4 procedures (*Hormigo et al., 2021b*). During AA3, animals learn to discriminate between two CSs, passively avoiding when AA3-CS2 is presented. *Figure 6C* shows $F/F_o$ and movement traces from CS onset for the AA1 (black), AA2 (red), and AA3 (green) procedures classified as correct (left panel, avoids; for AA3-CS2 correct passive avoids are shown in blue) or incorrect responses (right panel, escapes). During AA1, the CS caused a very small but noticeable $F/F_o$ peak at CS onset (*Figure 6C*) that is associated with a typical orienting head movement that occurs at CS onset depending on task contingencies and SPL (*Zhou et al., 2023*). However, this early $F/F_o$ peak was small and barely noticeable in most animals as was the averaged orienting movement caused by the CS at this SPL. In the absence of orienting movement, the tone did not evoke a noticeable activation of zona incerta neurons at CS onset. This differs from other neurons subjected to the same CS, such as superior colliculus and pedunculo-pontine tegmentum (PPT) neurons, which activate strongly at CS onset as reported in previous studies using the same methods (*Hormigo et al., 2021a*; *Zhou et al., 2023*).

While CS onset produced little activation, the succeeding avoidance movement was associated with strong zona incerta neuron activation (*Figure 6C* avoids from CS onset; note that since avoids occur at different latencies after CS onset they are best measured from their occurrence as in *Figure 6D*). In AA1, there was a large $F/F_o$ peak that followed the avoidance movement. As mice transitioned to AA2, the zona incerta activation shifted to the right following the delayed avoidance latencies characteristic of this procedure (*Figure 6C*, black vs red traces). Thus, the activation of zona incerta neurons is closely associated with the active avoidance movement. In addition, when mice failed to avoid, zona incerta neurons activated very sharply in association with the fast escape responses evoked by the US during the escape interval (*Figure 6C*, escapes). Furthermore, during the AA3 procedure, only CS1, which drives active avoidance responses, produced strong zona incerta activation (*Figure 6C*, green). CS2, which drives passive avoidance, produced nil activation and movement (*Figure 6C*, blue). To measure avoidance and escape responses, we extracted the $F/F_o$ and speed from response occurrence (*Figure 6D*). Both avoids and escapes were associated with robust zona incerta activation during the three procedures (*Figure 6D, E*). Moreover, escapes were faster (Tukey $t(7)$ = 7.53, p = 0.0011) and produced stronger activation than avoids (Tukey $t(7)$ = 5.44, p = 0.0063) for AA1, AA2, and AA3.

During AA4, mice adapt their avoidance movement to the duration of the avoidance interval signaled by each of the three CSs (*Hormigo et al., 2021b*). Accordingly, zona incerta activation shifted to reflect the avoidance movement (*Figure 7A*). When responses were measured from response occurrence (*Figure 7A*, right panel and *Figure 7B*), the CS1 avoids were faster than CS2 (Tukey $t(10)$ = 6.78, p = 0.0019) or CS3 avoids (Tukey $t(10)$ = 5.86, p = 0.0051), consistent with the more imminent threat signaled by CS1, which has a shorter 4-s avoidance interval. However, this was not reflected in a difference of the peak activation of zona incerta neurons (RMAnova $F(2,10)$ = 0.09, p = 0.91) perhaps because the slow $F/F_o$ measure cannot discriminate such a subtle difference in speed. In conclusion, zona incerta activation was closely associated with movement during the tasks, not with the behavioral tasks being performed.

Since mice have high rates of avoidance responses in these tasks, and consequently relatively few escape responses, we conducted additional sessions (*Figure 7C, D*; seven mice) in which the US (foot-shock and white noise) was presented unsignaled. The unsignaled US drives escape responses on every trial. To distinguish the contribution of the foot-shock and white noise presented by the US, additional trials in the same session presented the foot-shock or the white noise alone. The unsignaled US evoked a strong zona incerta activation in association with the fast escape response. Presentation of the foot-shock alone, produced an activation and escape response that was similar to the full US. Presentation of the white noise alone also drove escape responses, but these were somewhat slower

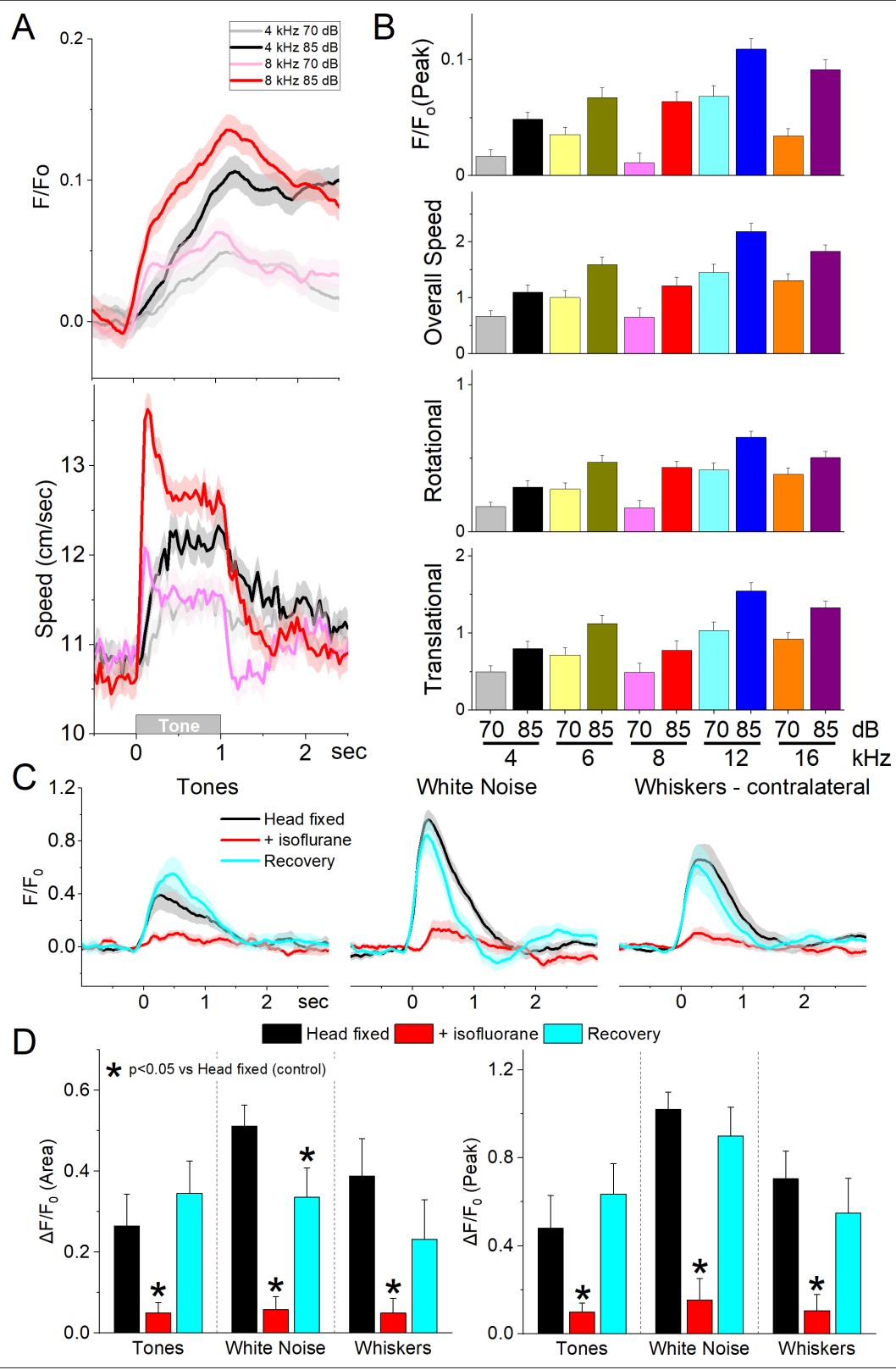

**Figure 5.** GABAergic zona incerta neurons discharge to auditory tones in association with movement. (**A**) Example $F/F_o$ calcium imaging and movement traces (mean ± standard error of the mean [SEM]) evoked from zona incerta neurons by auditory tones (1 s) of different saliency. The tones vary in frequency (kHz) and SPL (dB). (**B**) Area of $F/F_o$ and movement components measured during a time window (0–2 s) after tone onset. (**C**) $F/F_o$ responses of zona

*Figure 5 continued on next page*

*Figure 5 continued*

incerta GABAergic neurons evoked in head-fixed lightly restrained mice by auditory tones (1 s), while noise (1 s), and air-puffs (20 ms) aimed at the contralateral whiskers as mice transition between waking (head fixed), isoflurane anesthesia and recovery from anesthesia. (**D**) Population measurements of area (0–2 s) and peak amplitude from pre-stimulus baseline for the conditions in C.

and evoked less zona incerta activation than foot-shock alone (Tukey $t(6)$ = 5.42, p = 0.02) or the full US (Tukey $t(6)$ = 5.69, p = 0.01). Thus, the zona incerta activation during unsignaled US presentations, follows the speed of the escape movement driven by the stimuli, and the auditory white noise component adds little to the foot-shock driven movement and its related activation. In conclusion, zona incerta neurons activate to stimuli that produce movement.

## Zona incerta neuron lesions have little effect on goal-directed movement

The previous results show that GABAergic zona incerta neurons activate robustly during movement in various goal-directed avoidance behaviors. To determine if these neurons are important for learning and/or performing avoidance behaviors, we lesioned GABAergic zona incerta neurons by bilaterally injecting AAV8-EF1a-mCherry-flex-dtA (Neurophotonics) into the zona incerta of Vgat-cre mice. To verify the lesion, we counted the number of neurons (Neurotrace) in the zona incerta lesion and control mice. The injection reduced the number of zona incerta neurons (*Figure 8A*, Mann–Whitney $Z$ = 4.09, p<0.0001 Lesion vs Control). However, killing GABAergic zona incerta neurons had only minor effects on the ability of mice to learn and perform signaled active avoidance tasks (*Figure 8B*). There was a small effect on the percentage of avoidance responses during the AA1 procedure (ANOVA $F(1,13)$ = 4.89, p = 0.043, Lesion vs Control) but not during AA2 or AA3. There was no difference in either avoidance latency or ITCs between control and lesion mice.

Comparison of movement during task performance revealed significant differences between lesion and control mice. Since these effects were similar in the AA1, AA2, and AA3 tasks, we combined the results for the three procedures (*Figure 8C, D*). The orienting response evoked by the CS1 onset was larger in lesion mice compared to the control mice (ANOVA $F(1,343)$ = 64.72, p < 0.0001). However, the avoidance movement speed was slower in the lesion mice compared to the control mice (ANOVA $F(1,335)$ = 37.95, p < 0.0001). This applied to both the translational (ANOVA $F(1,335)$ = 44.65, p < 0.0001) and rotational (ANOVA $F(1,335)$ = 8.44, p = 0.0039) movement. Similarly, escape response movement was also slower in lesion mice compared to control mice (ANOVA $F(1,250)$ = 8.9, p = 0.0031). This applied to translational (ANOVA $F(1,250)$ = 12.16, p = 0.0006) but not rotational (ANOVA $F(1,250)$ = 1.04, p = 0.3) movement. In conclusion, lesions of GABAergic zona incerta neurons enhance the CS-evoked orienting reflex but suppress the goal-directed movement driven by the CS and US. The lesions have little impact on brain processes such as decision making (Go/NoGo in AA1–3), response inhibition (unsignaled and signaled passive avoidance in AA2/3), and stimulus discrimination (AA3).

## Effects of zona incerta inhibition and excitation on exploratory and goal-directed movement

Since zona incerta neurons activate in association with movement during exploratory and goal-directed behaviors, we determined the effects of manipulating the activity of zona incerta neurons unilaterally and bilaterally on these behaviors. We used optogenetics to inhibit or excite GABAergic zona incerta neurons, which we previously validated with slice and in vivo recordings, including zona incerta GABAergic neurons during signaled avoidance (*Hormigo et al., 2016*; *Hormigo et al., 2019*; *Hormigo et al., 2021a*; *Hormigo et al., 2021c*). The optogenetic groups were compared to themselves (control vs optogenetic trials in the same sessions) and to a No-Opsin group that did not express any opsins but was subjected to the same light protocols. In general, application of continuous (Cont) blue or green light, or trains (1-ms pulses) of blue light at 10–20 or 40–66 Hz in the zona incerta of No Opsin mice (27 sessions in 7 mice; blue and green light are combined in Cont) while they explored an open field did not affect speed or head direction bias in an open field (*Figure 9A, B*).

Unilateral inhibition of Arch-expressing GABAergic zona incerta neurons in Vgat-ZI-Arch mice (35 sessions in 7 mice; *Figure 9A–C*) with Cont green light did not produce a significant effect on

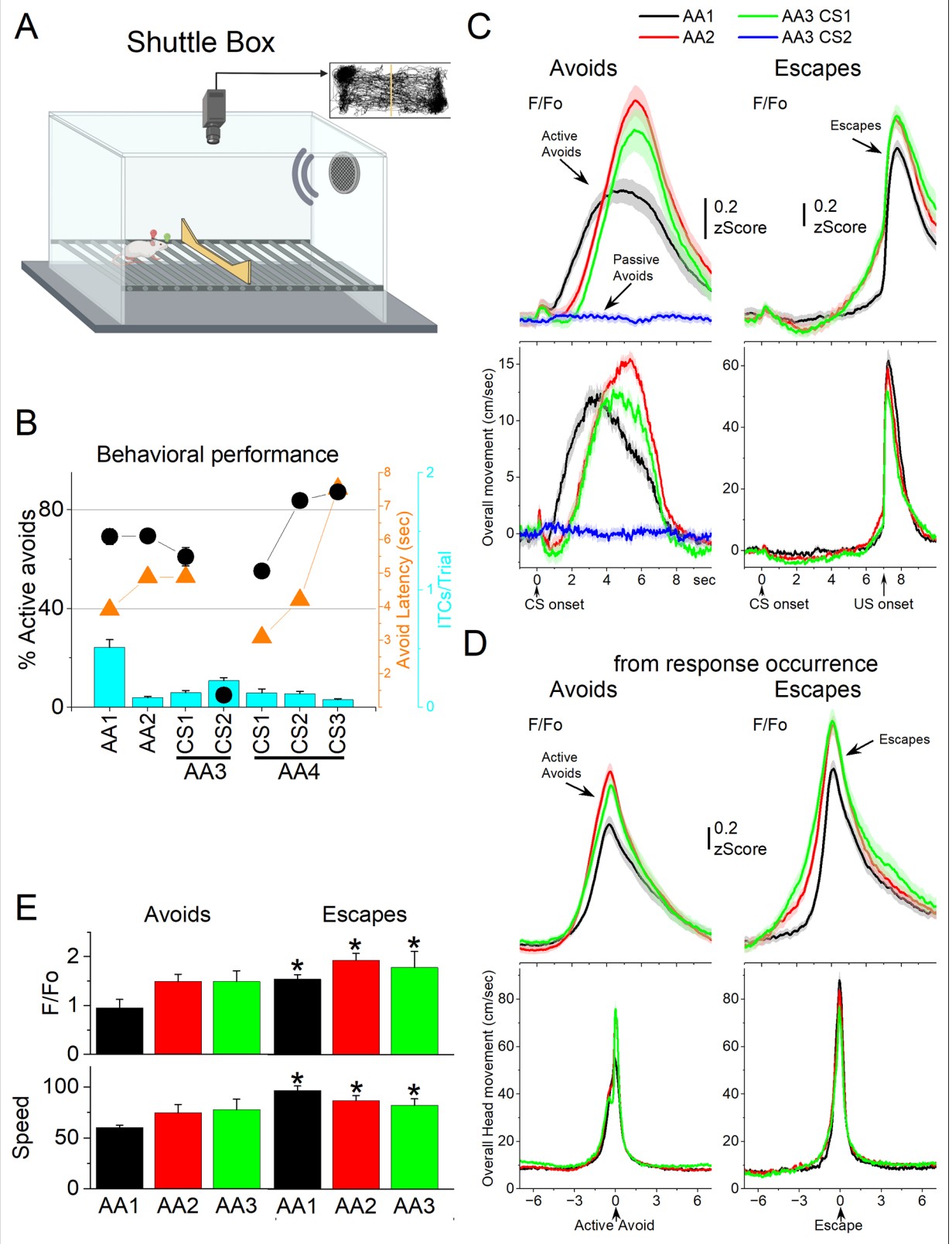

**Figure 6.** GABAergic zona incerta activation in the context of signaled active avoidance. (**A**) Arrangement of the shuttle box used during signaled avoidance tasks. (**B**) Behavioral performance during the four different avoidance procedures showing the percentage of active avoids (black circles), avoidance latency (orange triangles), and intertrial crossings (ITCs) (cyan bars). (**C**) $F/F_o$ and overall movement traces from CS onset for AA1, AA2, and AA3 (CS1 and CS2) procedures per trials classified as avoids (left) or escapes (right) of CS-evoked orienting response measured by tracking overall head

*Figure 6 continued on next page*

*Figure 6 continued*

speed. (**D**) Same as C from response occurrence. (**E**) Population measures of $F/F_o$ and speed for avoids and escapes during AA1, AA2, and AA3 (CS1). Asterisks denote p < 0.05 versus Avoids.

head orienting bias compared to No Opsin controls (Cont vs No Opsin Tukey $t(68)$ = 1.64404, p = 0.24915 [p = 0.218]). There was a small tendency for the head to move contraversively after light onset followed by an ipsiversive rebound at light offset. These effects were small (<5°) but produced a significant increase in peak speed only during the ipsiversive rebound after light offset. Thus, inhibiting GABAergic zona incerta cells unilaterally has weak effects on head orienting movement.

Bilateral inhibition of zona incerta was tested during performance of signaled active avoidance (AA1). Inhibiting Arch-expressing GABAergic zona incerta neurons in Vgat-ZI-Arch mice (21 sessions in 7 mice; *Figure 9D*) with Cont green light (during the CS–US periods) increased the overall speed of the orienting response movement (Tukey $t(6)$ = 8.57, p = 0.0009 area; Tukey $t(6)$ = 7.63, p = 0.0017 peak) and increased the ensuing avoidance movement (Tukey $t(6)$ = 5.97, p = 0.0056 area) shifting the time to peak of avoidance faster (Tukey $t(6)$ = 8.47, p = 0.001) without increasing the peak avoidance speed (Tukey $t(6)$ = 0.01, p = 0.995). This was reflected in both the translational and rotational components of the avoidance movement. As previously shown, the effects on movement caused by zona incerta inhibition slightly improved avoidance performance but these effects are small because mice already have high levels of performance (*Hormigo et al., 2020*). Thus, inhibiting zona incerta neurons enhances goal-directed movement and the orienting response driven by the CS.

Next, we tested the effect of unilaterally exciting zona incerta GABAergic neurons on exploratory and goal-directed behaviors. Excitation of ChR2-expressing GABAergic zona incerta neurons in Vgat-ZI-ChR2 mice (44 sessions in 4 mice; *Figure 10A, C*) with Cont and 40–66 Hz blue light evoked a contraversive orienting head bias compared to No Opsin controls (Cont vs No Opsin Tukey $t(156)$ = 15.48, p < 0.0001 [p < 0.0001]; 40–66 Hz vs No Opsin Tukey $t(156)$ = 9.67, p < 0.0001 [p < 0.0001]). The strongest movement was caused by Cont blue light, which produced a sharp change in movement vigor at light onset followed by a smaller rebound at light offset. However, the effect of unilateral excitation on movement vigor was virtually absent during bilateral excitation (*Figure 10B, C*). Thus, during exploratory behavior, unilateral excitation of zona incerta GABAergic neurons evokes a vigorous contraversive movement. However, bilateral excitation was not associated with movement or directional bias.

Bilateral excitation of zona incerta GABAergic neurons during performance of signaled active avoidance (AA1; *Figure 10D*) increased the overall (baseline subtracted) speed of the orienting response movement (Tukey $t(28)$ = 5.64, p = 0.0004 area; Tukey $t(28)$ = 4.19, p = 0.0062 peak) and decreased the ensuing avoidance movement (Tukey $t(28)$ = 3.57, p = 0.0176 area; Tukey $t(28)$ = 6.21, p = 0.0001 peak). As previously shown, the effects on movement caused by zona incerta excitation can severely impair avoidance performance as a function of the level of activation through projections to the midbrain PPT (*Hormigo et al., 2020*). However, these levels of excitation imposed by optogenetics may represent abnormal states that are not expressed during normal zona incerta activation. Thus, exciting GABAergic zona incerta cells bilaterally suppresses avoidance movement which can impair goal-directed behavior.

## What targets mediate the effects of exciting zona incerta on exploratory movement?

Excitation of the projection fibers of zona incerta GABAergic neurons in different target locations produced distinct effects. In the PO thalamus, unilateral excitation of ChR2-expressing GABAergic fibers originating in zona incerta (31 sessions in 4 mice; *Figure 11A, C, D*) with Cont and 40–66 Hz blue light evoked a contraversive orienting head bias compared to No Opsin controls (Cont vs No Opsin Tukey $t(130)$ = 7.15, p < 0.0001 [p < 0.0001]; 40–66 Hz vs No Opsin Tukey $t(130)$ = 4.16, p < 0.0001 [p < 0.0001]). The strongest movement was caused by unilateral Cont blue light, which produced a sharp change in movement vigor at light onset. The effects of unilateral Cont blue light in PO resemble the effects of light applied in zona incerta proper. This is expected because we previously showed that Cont light applied in PO thalamus can reach the underlying zona incerta (*Hormigo et al., 2020*). In contrast, the shorter 1 ms trains, which are much more localized, produced weaker effects, albeit in the same direction. However, none of these effects were observed during bilateral

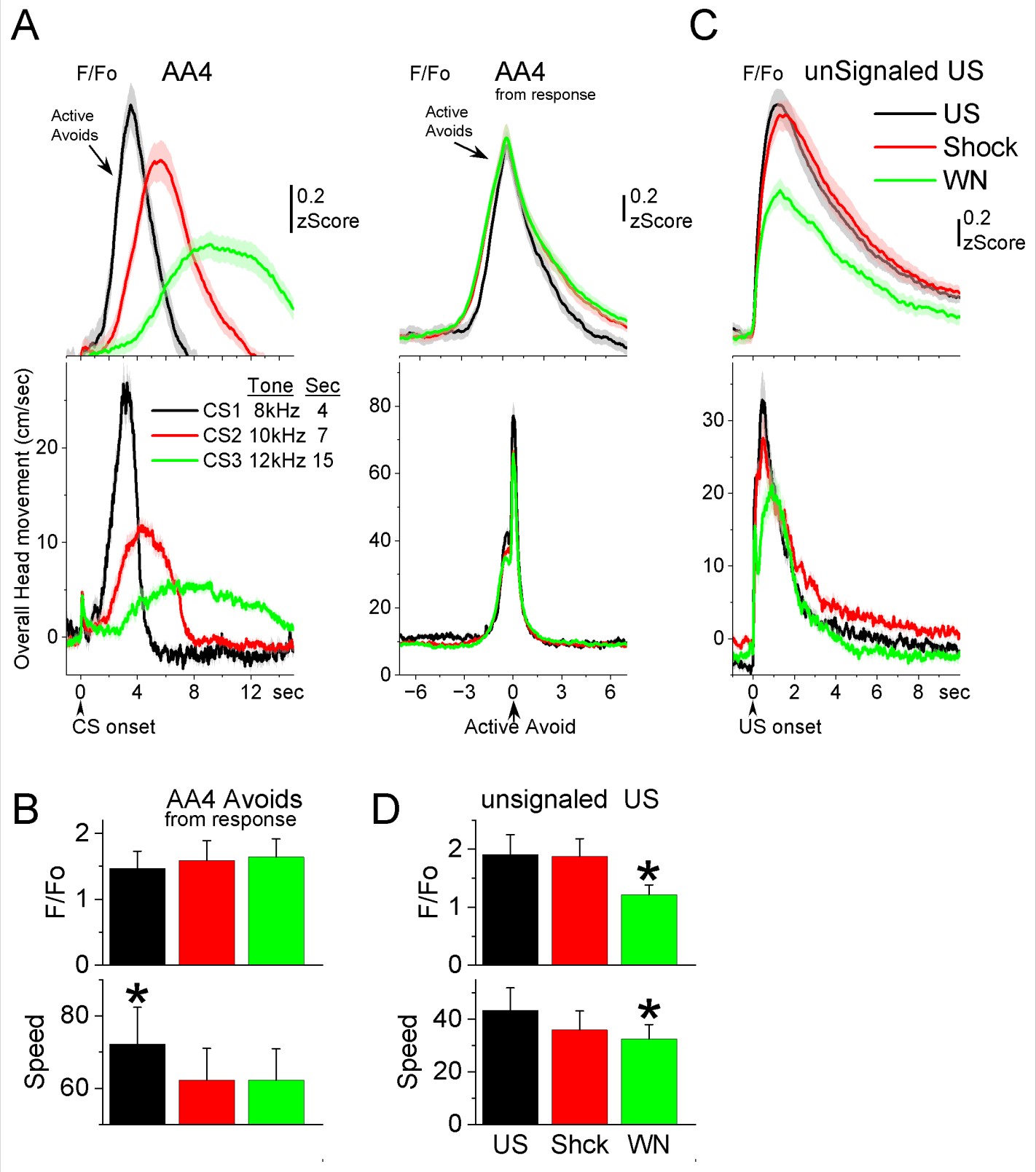

**Figure 7.** GABAergic zona incerta neurons track the avoidance and escape movement. (**A**) $F/F_o$ and overall movement traces from CS onset (left) and response occurrence (right) for avoids during the AA4 procedure, which include three CSs that signal avoidance intervals of different durations. (**B**) Population measures (−3 to 3 s area, mean ± standard error of the mean [SEM]) from response occurrence. Asterisks denote significant differences versus other stimuli. (**C**) $F/F_o$ and overall movement traces from unconditioned stimulus (US) onset for escapes during the unsignaled US procedure,

*Figure 7 continued on next page*

*Figure 7 continued*

which includes the US, or each of its components delivered alone (foot-shock and white noise). (**D**) Population measures (0–5 s area, mean ± SEM) for the data in C. Asterisks denote significant differences versus other CSs.

excitation (*Figure 11B–D*), which is also the case when light is applied directly in zona incerta. Thus, unilateral PO inhibition caused by zona incerta firing slightly biases head orienting in the contraversive direction while bilateral PO inhibition has little effect on movement.

In PPT, unilateral excitation of ChR2-expressing GABAergic fibers originating in zona incerta (34 sessions in 4 mice; *Figure 11A, C, D*) with 40–66 Hz trains of blue light, but not Cont, evoked a contraversive orienting head bias compared to No Opsin controls (Cont vs No Opsin Tukey $t(136)$ = 2.6, p = 0.42 [p = 0.68]; 40–66 Hz vs No Opsin Tukey $t(136)$ = 11.23, p < 0.0001 [p < 0.0001]). The movement was caused selectively by trains of blue light, which produced a change in movement vigor at light onset. However, these effects of movement were absent when the excitation was bilateral (*Figure 11B–D*). Thus, only unilateral PPT inhibition by zona incerta fibers biases head movement.

In contrast to the effects within zona incerta or projections to PO and PPT, in the superior colliculus, unilateral excitation of ChR2-expressing GABAergic fibers originating in zona incerta (23 sessions in 4 mice; *Figure 11A, C, D*) with Cont and 40–66 Hz blue light evoked an ipsiversive orienting head bias compared to No Opsin controls (Cont vs No Opsin Tukey $t(114)$ = 5.58, p = 0.001 [p < 0.0001]; 40–66 Hz vs No Opsin Tukey $t(114)$ = 3.75, p = 0.09 [p = 0.002]). The strongest movement was caused by unilateral Cont blue light, which produced a change in movement vigor at light onset. However, these effects were not observed during bilateral stimulation (*Figure 11B–D*). The effects of unilateral excitation of zona incerta fibers in superior colliculus, which inhibits the superior colliculus, are consistent with the effects of unilateral superior colliculus inhibition with other methods (*Zhou et al., 2023*).

In conclusion, inhibition of zona incerta GABAergic neurons has weak effects on exploratory movement. In contrast, unilateral excitation of zona incerta GABAergic neurons produces a contraversive head orienting bias associated with a sharp increase in movement vigor at light onset that is absent during bilateral excitation. The unilateral effects of excitation can be mediated primarily by projections to PPT in the midbrain, but less likely by projections to PO thalamus or superior colliculus, which produce weak effects on movement or movement in the ipsiversive (opposite) direction, respectively. The fact that unilateral excitation of GABAergic pathways to the midbrain tegmentum and to the superior colliculus have opposite effects on movement orienting direction agrees with the finding that population activity of zona incerta GABAergic neurons measured with fiber photometry do not code orienting direction.

## Discussion

We found that zona incerta neurons provide a generalized movement signal to the distributed network of sites where they project. Electrophysiology revealed that most zona incerta neurons are modulated by movement with some neurons strongly activated and others deactivated by movement. Calcium imaging showed that GABAergic zona incerta neurons activate during exploratory and goal-directed movement, but they do not code the direction of the movement. Although zona incerta GABAergic neurons activate during goal-directed behaviors, removing (inhibiting or killing) these neurons had little impact on behavioral performance. Removing these neurons modulated movement parameters, such as goal-directed speed and CS-evoked orienting responses, which highlights a role of zona incerta in sensorimotor regulation, but animals performed the tasks at control levels.

Although optogenetic excitation of zona incerta GABAergic neurons can suppress goal-directed movement and impact behavioral performance (*Hormigo et al., 2020*), this may represent an abnormal level of activation associated with pathological states. In other words, excitation of a pathway that inhibits the circuits responsible for generating a behavior will inhibit that behavior regardless of whether this occurs normally. Indeed, zona incerta projects to the midbrain pedunculopontine tegmentum, a region that is critical for the avoidance goal-directed behavior we studied (*Hormigo et al., 2019*), and accordingly exciting this pathway suppresses the behavior (*Hormigo et al., 2020*). Furthermore, when we recorded from these neurons, we found a robust activation during goal-directed movement, which emphasizes the point that the levels of optogenetic excitation that suppress behavioral performance are likely distinct from the levels of activation associated with normal movement. Since the removal of

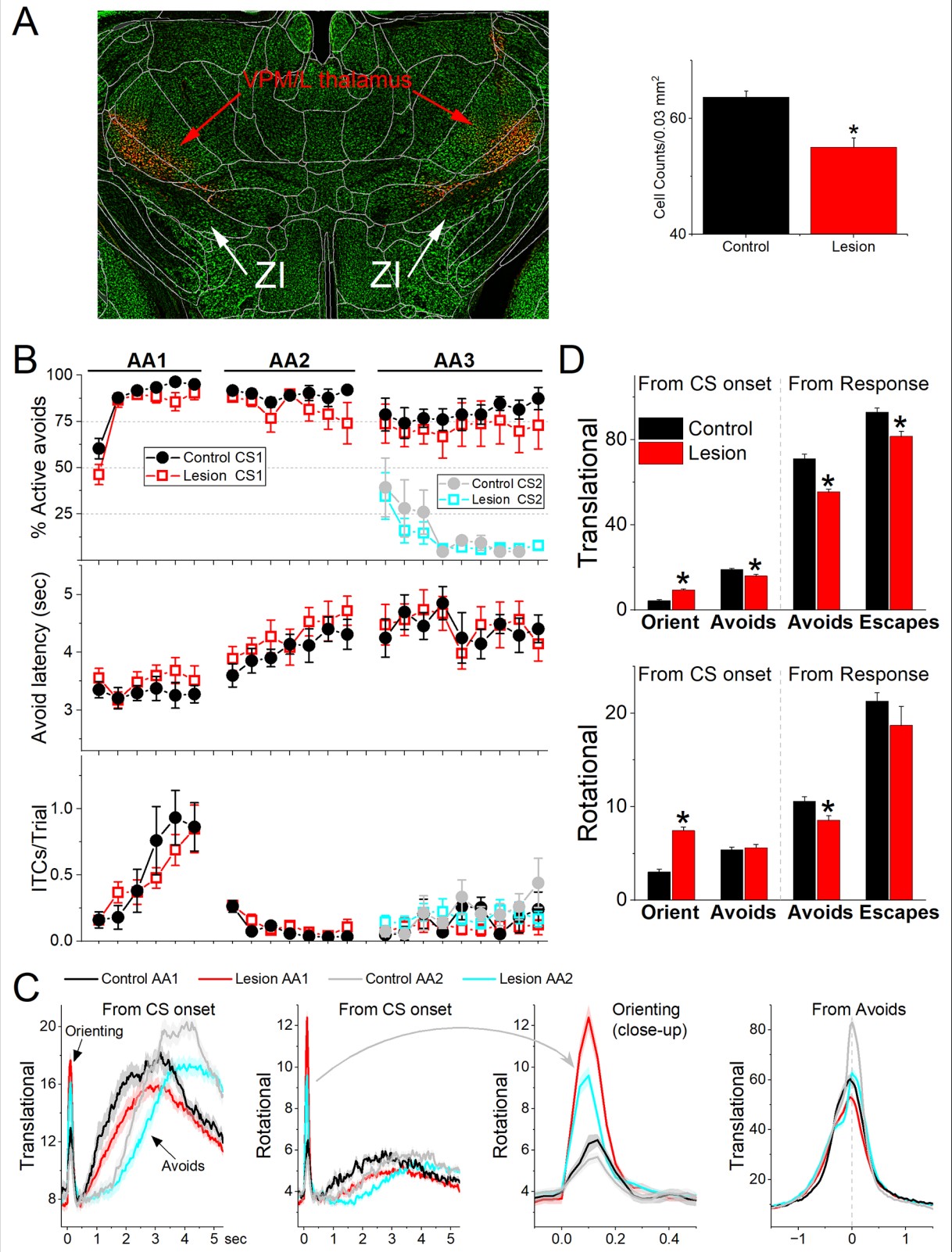

**Figure 8.** Lesions of GABAergic zona incerta neurons disrupt movement but does not abolish signaled active avoidance learning or performance. (**A**) Coronal Neurotrace (green) strained section of a Vgat-Cre mouse injected with a Cre-depedent AAV-dtA in the zona incerta to kill GABAergic neurons. The red fluorophore overlying and around zona incerta is expressed in non-Cre cells signifying a successful injection. We counted the number of cells in the zona incerta in controls and injured mice and found a significant reduction of zona incerta neurons. (**B**) Behavioral performance during AA1, AA2,

*Figure 8 continued on next page*

*Figure 8 continued*

and AA3 procedures showing the percentage of active avoids (upper), avoid latency (middle), and intertrial crossings (ITCs) (lower) for control and lesion mice. The AA3 procedure shows CS1 and CS2 trials for the same sessions. Note that active avoids during AA3-CS2 trials are errors, as the mice must passively avoid during CS2. (**C**) Translational and rotational movement from CS onset during AA1 and AA2 procedures for control and lesion mice. A closeup of the orienting response evoked by the CS for the rotational movement is shown in the third panel. (**D**) Population measures of orienting, avoidance, and escape responses from CS onset and from response occurrence for translational and rotational movement. Asterisks denote p < 0.05 versus controls.

this activation has little impact on behavioral performance, the role of zona incerta activation cannot be to drive the behavior but to inform its targets that the behavior is occurring, which can be used to modulate the behavior.

## Zona incerta and sensorimotor processing

Consistent with the existence of auditory inputs located laterally and somatosensory inputs distributed throughout zona incerta in rats (*Mitrofanis, 2002*; *Shaw and Mitrofanis, 2002*), previous studies have shown that zona incerta neurons discharge in response to auditory and whisker stimulation under various conditions (*Trageser et al., 2006*; *Urbain and Deschênes, 2007*; *Wang et al., 2019*; *Zhao et al., 2019*). Thus, a sensory integrative function has been proposed as its main role (*Mitrofanis, 2005*; *Wang et al., 2020a*; *Fratzl and Hofer, 2022*). We found that during auditory mapping sessions, zona incerta GABAergic neurons discharged in response to auditory stimuli that did not predict any behavioral contingencies. However, the auditory stimuli that activated zona incerta neurons were invariably associated with movement. While head fixation and mild restraint did not abolish zona incerta activations evoked by sensory (auditory and whisker) stimuli, this also did not abolish all movements associated with presentation of the stimuli, such as pinna, whisker, facial, and limb movements that are typical under these conditions. Light isoflurane anesthesia was required to abolish the evoked movements, which also strongly suppressed or abolished the sensory-evoked responses in zona incerta. However, as already noted, the suppression of sensory responses may be due to changes in arousal (*Castro-Alamancos, 2004*; *Lee and Dan, 2012*) and not caused by the abolishment of the movements per se. Taken together, the results indicate that zona incerta GABAergic neurons respond to salient auditory stimuli primarily in association with movement.

The orienting reflex is a basic sensorimotor behavior that consists of a fast (<300 ms) head movement in association with presentation of sensory stimuli (*Sokolov, 1963*), which is modulated by behavioral states (*Zhou et al., 2023*). Intriguingly, lesions of zona incerta GABAergic neurons produced a large increase of the orienting response evoked by the CS in the context of signaled goal-directed avoidance behavior. Moreover, the enhanced orienting reflex contrasted with the suppression of the goal-directed movement in the same lesion animals. Similarly, optogenetic inhibition or excitation of zona incerta GABAergic neurons augmented the CS-evoked orienting responses in the same context. Thus, the zona incerta has access to areas that control the orienting reflex. Indeed, zona incerta projections to midbrain areas, such as the superior colliculus (*Zhou et al., 2023*), can easily mediate these effects. One possibility is that the brainstem nuclei responsible for the orienting reflex are under tonic zona incerta inhibitory control and removing or dampening this inhibition augments the orienting reflex. Since zona incerta can modulate the orienting reflex, it must be considered as part of the circuitry that may control this response during different behavioral states.

## Zona incerta activates during goal-directed movement

The most robust activation of zona incerta neurons is associated with movement. In head-fixed mice on the spherical treadmill, we found that zona incerta neurons discharge in relation to the occurrence of movement. Two major groups of neurons had different coding of movement and could be further classified into subgroups. Class1 neurons were activated with movement and could be differentiated based on their firing rate, with Class1a having higher firing rates than Class1b. Class2 neurons consisted of two distinct subclasses, with Class2a neurons somewhat deactivated during movement, and Class2b neurons poorly responsive to movement. Most of the tested units did not respond to auditory stimuli perhaps because our unit sampling missed the lateral parts of zona incerta, which receive auditory input in rats (*Mitrofanis, 2002*). A small percentage of neurons responded to whisker

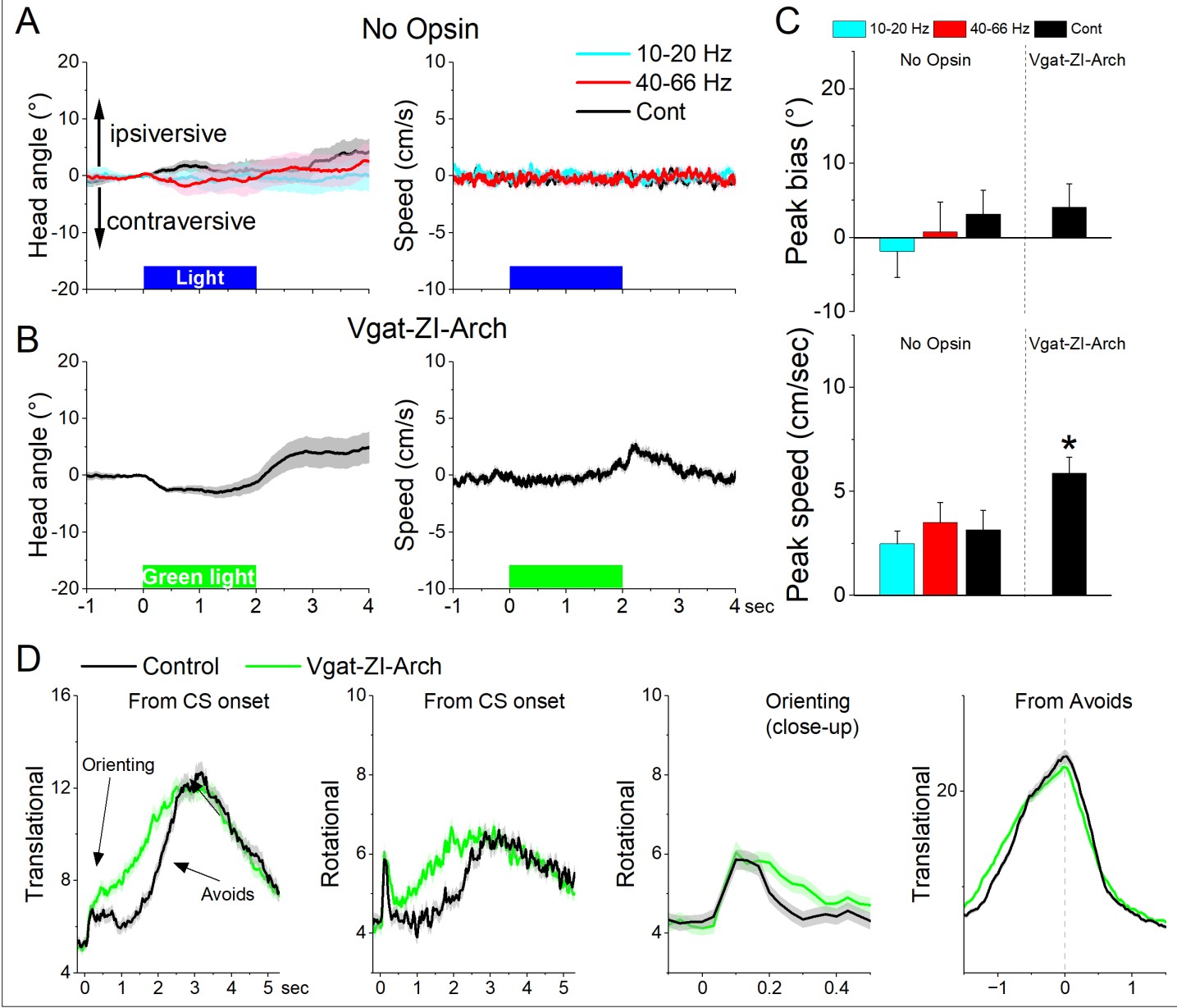

**Figure 9.** Optogenetic inhibition of GABAergic zona incerta neurons during exploratory and goal-directed movements. (**A**) Effects of unilateral trains (blue only) or continuous (blue or green) light on head bias (angle in degrees), and speed movement (cm/s) during exploration of an open field for No Opsin controls. Negative values represent contraversive turns versus the side of the unilateral light. (**B**) Same as A for Vgat-ZI-Arch mice subjected to continuous green light pulses that inhibit zona incerta neurons. (**C**) Population measures of peak head bias and movement speed for the groups in A and B. Asterisks denote p < 0.05 versus No Opsin mice. (**D**) Translational and rotational movement components (speed) for Vgat-ZI-Arch mice performing AA1. The traces are from CS onset or from avoid occurrence for trials subjected to continuous bilateral green light (green) or control trials (black) delivered randomly during the same session. The third panel is a closeup of the second panel depicting the orienting response. The light is presented during the avoidance/escape intervals. All traces in the figure are mean ± standard error of the mean (SEM).

stimulation, but these neurons belonged to both Class1 and Class2, indicating that they were not associated with a particular sensitivity to movement.

In agreement with the unit data, fiber photometry calcium imaging of zona incerta GABAergic neurons revealed a robust zona incerta neuron activation during exploratory movement. Exploratory movement is characterized by orienting turns of the head as the animal samples the environment (*Wilson et al., 2018*; *Masullo et al., 2019*). However, the robust activation of these neurons did not code the orienting direction of the exploratory movement. Zona incerta neurons discharge during

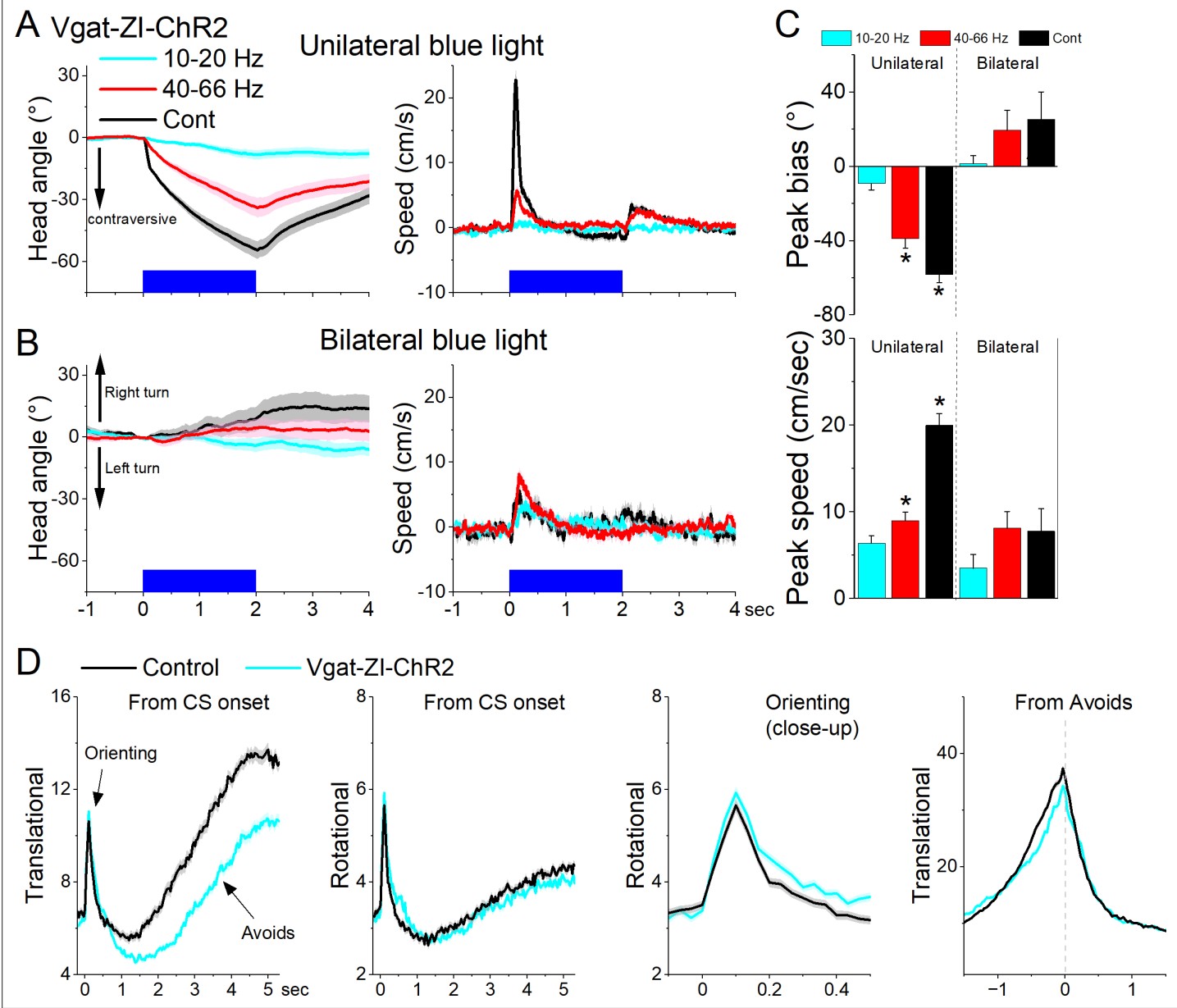

**Figure 10.** Optogenetic excitation of GABAergic zona incerta neurons during exploratory and goal-directed movements. Effects of unilateral (**A**) or bilateral (**B**) trains or continuous blue light on head bias (angle in degrees), and speed (cm/s) during exploration of an open field for Vgat-ZI-ChR2 mice. Unilateral Cont and high-frequency trains of blue light drove vigorous contraversive head movements. Bilateral stimulation did not drive vigorous head movement. During unilateral light stimulation, negative values represent contraversive turns versus the side of the unilateral light. During bilateral light stimulation, negative values represent left turns. (**C**) Population measures of peak head bias and speed for the groups in A and B. Asterisks denote $p < 0.05$ versus No Opsin mice. (**D**), Translational and rotational movement components (speed) for Vgat-ZI-ChR2 mice performing AA1. The traces are from CS onset or from avoid occurrence for trials subjected to continuous bilateral green light (green) or control trials (black) delivered randomly during the same session. The third panel is a closeup of the second panel depicting the orienting response. The light is presented during the avoidance/escape intervals. All traces in the figure are mean ± standard error of the mean (SEM).

movement occurrence regardless of the animal's turning direction, which contrasts with one of its targets, the superior colliculus, that clearly codes the orienting direction using the same recording methods (*Zhou et al., 2023*). This indicates that zona incerta neurons provide a generalized movement signal to the distributed network of sites where they project.

We used several procedures to test the relationship between goal-directed behavior and zona incerta neuron activity. Zona incerta neurons were strongly activated during avoidance and escape

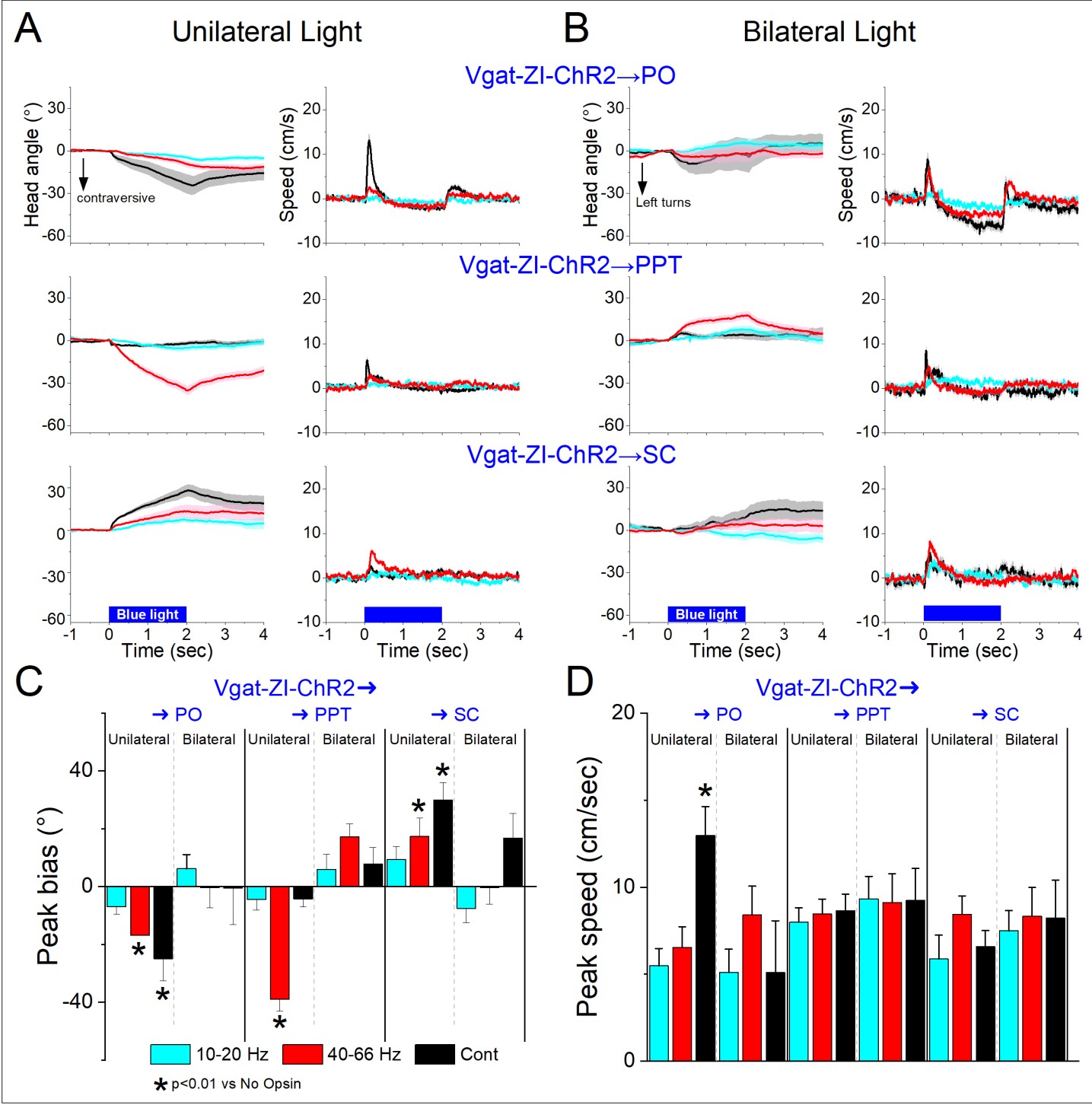

**Figure 11.** Optogenetic excitation of GABAergic zona incerta pathways to PO, pedunculo-pontine tegmentum (PPT), and superior colliculus during exploratory movements. Effects of unilateral (**A**) or bilateral (**B**) trains or continuous blue light on head bias (angle in degrees), and speed (cm/s) during exploration of an open field for Vgat-ZI-ChR2 → PO, Vgat-ZI-ChR2 → PPT, and Vgat-ZI-ChR2 → SC mice. During unilateral light stimulation, negative values represent contraversive turns versus the side of the unilateral light. During bilateral light stimulation, negative values represent left turns. Population measures of peak head bias (**C**) and speed (**D**) for the groups in A/B. Asterisks denote $p < 0.05$ versus No Opsin mice. All traces in the figure are mean ± standard error of the mean (SEM).

movements. Passive avoids driven by auditory stimuli, which are not associated with movement, did not activate zona incerta neurons. Furthermore, unsignaled US presentations revealed a strong zona incerta activation during fast escape responses. Dissociation of the US components (foot-shock and auditory white noise) showed that the auditory stimulus adds little to the foot-shock-evoked activation. Thus, zona incerta neurons activate in response to stimuli that produce movement regardless of the sensory modality driving the movement. Moreover, although excitation of zona incerta GABAergic neurons can suppress active avoids, zona incerta does not cause the suppression of movement associated with passive avoids because there was little activation of these neurons during signaled passive avoidance trials (CS2 in AA3). This emphasizes the point that although the circuit can generate a certain effect during optogenetic stimulation, it does not mean that it normally produces that effect during normal operations.

Our goal-directed tasks engage a variety of neural processes including decision making (Go/NoGo in AA1–3), action inhibition (unsignaled and signaled passive avoidance in AA2–3), and stimulus discrimination (AA3). Lesions or optogenetic inhibition of zona incerta GABAergic neurons affected movement speed during the goal-directed behaviors but had little impact on the ability of mice to successfully perform the tasks. Moreover, this regulation of speed was bidirectional; excitation of zona incerta suppresses goal-directed movement while inhibition of zona incerta facilitates goal-directed movement. This seems consistent with a role in modulating speed as a function of zona incerta activation. For instance, if zona incerta activation is strong it would inhibit the brainstem targets that generate the ongoing movement to keep it under control (within certain parameters), as part of a regulatory control system (*Sommer and Wurtz, 2008*; *Yin, 2017*). Thus, the zona incerta would function as a sensor of ongoing movement that feeds control circuits in the brainstem through its projections to keep the behavior within certain parameters defined by the situation. We conclude that an important role of zona incerta neurons is to signal to target structures that movement is occurring, which is useful to regulate ongoing behavior.

# Materials and methods
## Experimental design and statistical analysis

All procedures were reviewed and approved by the institutional animal care and use committee and conducted in adult (>8 weeks) male and female mice. Experiments involved a repeated measures design in which the mice or cells serve as their own controls (comparisons within groups), but we also compared experimental groups of mice or cells between groups (comparisons between groups). For comparisons within groups, we tested for a main effect of a variable (e.g., Stimulus) using a repeated measures ANOVA or a linear mixed-effects model with fixed-effects and random-effects (e.g., sessions nested within the Subjects as per Data ~Stimulus + (1|Subjects/Sessions lme4 syntax in R)) followed by comparisons with Tukey's test. For comparisons between different groups, we used the same approach but included the Group as an additional fixed-effect (Data ~Group*Stimulus + (1|Subjects/Sessions)). Using the standard errors derived from the model, Tukey tests were conducted for the effect of the fixed-effect factors (within group comparisons) or for the Group–Stimulus interaction (between group comparisons). We report the Tukey values for the relevant multiple comparisons. Experiments always include both biological (animal Subjects) and technical replicates (Sessions nested within the Subjects). In addition, in many cases, we performed the same comparisons (both within groups and between groups) using a bootstrap approach by randomly sampling with replacement (1000–10,000 times) from all the values (irrespective of the conditions or groups) and determining the probability that the difference (or a larger difference) between the conditions or groups occurs by chance (these p values are reported inside brackets, []). Group sizes were based on power analysis using the measured means difference variability from previous similar experiments. For example, for signaled avoidance sessions this revealed that three animals in which we conducted five identical daily sessions per animal (15 sessions) was sufficient to detect a ~20% change in avoidance rate with a power of 0.99 ($p < 0.05$). Then, the number of animals used was increased (usually doubled) above the bare minimum. Experiments are conducted 'blind' because the sessions run automatically using predefined protocols linked to unique animal identifiers; it is not possible to influence the outcome even if the researcher is aware of the animal group. In addition, the analyses from the raw data to final reported statistics are automated/coded, and reproducible on demand. Animals are selected

randomly from our breeding colony (for the strain used), and trials delivered in each experiment are always randomized. If animals are excluded from an experiment, this will be stated in Results where the analyses are described including the criterion used for exclusion.

To enable rigorous approaches, we maintain a centralized metadata system that logs all details about the experiments and is engaged for data analyses (*Castro-Alamancos, 2022*). Moreover, during daily behavioral sessions, computers run experiments automatically using preset parameters logged for reference during analysis. Analyses are performed using scripts that automate all aspects of data analysis from access to logged metadata and data files to population statistics and graph generation. Data and code used for analyses are available directly from our managed servers upon request.

## Strains and adeno-associated viruses

To target GABAergic zona incerta neurons, we employed Vgat-Cre mice (breeding pairs were obtained from Jax stock 028862, bred in house as homozygotes, and we used the F1 generation at 1.5–3 months old of both sexes), where Cre-based expression selectively delineates the nucleus with nil expression in most areas immediately above (sensory ventral thalamus) or below (subthalamic nucleus) (*Vong et al., 2011*; *Hormigo et al., 2020*), which consist of glutamatergic neurons.

To record from GABAergic zona incerta neurons using calcium imaging, we injected a Cre-dependent adeno-associated virus (AAV) (AAV5-syn-FLEX-jGCaMP7f-WPRE (Addgene: $7 \times 10^{12}$ vg/ml)) in the zona incerta of Vgat-cre mice (Jax 028862; B6J.129S6(FVB)-Slc32a1$^{tm2(cre)Lowl}$/MwarJ) to express GCaMP6f/7f. An optical fiber was then placed in this location.

To knock-out Vgat from GABAergic zona incerta neurons, we injected AAV5.hSyn.HI.eGFP-Cre. WPRE.SV40 (UPenn vector core: $1.7 \times 10^{13}$ GC/ml) into the zona incerta of Vgat$^{flox/flox}$ mice (Jax 012897; Slc32a1$^{tm1Lowl}$/J). To kill GABAergic zona incerta neurons, we injected AAV8-EF1a-mCherry-flex-dtA (Neurophotonics: $1.3 \times 10^{13}$ GC/ml) into the zona incerta of Vgat-cre mice.

To inhibit GABAergic zona incerta neurons using optogenetics, we employed two groups of mice. In Vgat-ZI-Arch mice, we express Arch by injecting AAV5-EF1a-DIO-eArch3.0-EYFP (UNC Vector Core, titers: $3.4 \times 10^{12}$ vg/ml) in the zona incerta of Vgat-cre mice. In Vgat-ZI mice, we express Arch by crossing Vgat-cre and Ai40D (Jax 021188; B6.Cg-Gt(ROSA)26Sor$^{tm40.1(CAG-aop3/EGFP)Hze}$/J) mice.

To excite GABAergic zona incerta neurons using optogenetics, we employed Vgat-ZI-ChR2 mice in which we injected AAV5-EF1a-DIO-hChR2(H134R)-eYFP (UPenn Vector Core or Addgene, titers: $1.8 \times 10^{13}$ GC/ml by quantitative PCR) in the zona incerta of Vgat-cre mice.

For optogenetics, we implanted dual optical fibers bilaterally in the zona incerta (excitation and inhibition) or in zona incerta projection targets (excitation), including PO, superior colliculus, and the PPT. As No-Opsin controls, we injected AAV8-hSyn-EGFP (Addgene, titers: $4.3 \times 10^{12}$ GC/ml by quantitative PCR) or nil in the zona incerta. All the optogenetic methods used in the present study have been validated in previous studies using slice and/or in vivo electrophysiology (*Hormigo et al., 2016*; *Hormigo et al., 2019*; *Hormigo et al., 2021a*; *Hormigo et al., 2021c*).

## Surgeries

Optogenetics and fiber photometry experiments involved injecting 0.2–0.4 µl AAVs per site during isoflurane anesthesia (~1%). Animals received carprofen after surgery. The stereotaxic coordinates for injection in zona incerta are (in mm from bregma; lateral from the midline; ventral from the bregma-lambda plane): 2.3 posterior; 1.7; 3.7. In these experiments, a single (400 µm in diameter for fiber photometry) or dual (200 µm in diameter for optogenetics) optical fiber was implanted unilaterally or bilaterally during isoflurane anesthesia. The stereotaxic coordinates for the implanted optical fibers (in mm) are: zona incerta (2.3 posterior; 1.5; 3.8), PO thalamus (2.3 posterior; 1.5; 2.6), superior colliculus (4 posterior; 1–1.5; 1.5–1.8), and PPT (4.7 posterior; 1.25; 3.1 entering in the posterior direction at a 20° angle). The coordinate ranges reflect different animals that were combined because the slight coordinate differences produced similar effects.

## Fiber photometry

We employed a two-channel (465 and 405 nm) fiber photometry system (Doric Lenses) with alternating (20–60 and 20–50 µW, respectively) pulses of light excitation at 100 Hz (per each 10 ms, 465 is on for 3 ms, and 2 ms later 405 is on for 3 ms). The emission peak signals (525 and 430 nm for GCaMP6f and control emissions) evoked by the 465 nm and 405 pulses were acquired at 5–20 kHz

and measured at the end of each pulse. To calculate $F_o$, the 430 signal was scaled to the 525 signal ($F$) using the slope of the linear fit. Finally, $F/F_o$ was calculated with the following formula: $(F - F_o)/F_o$ and converted to Z-scores. Due to the nature of the behavior studied, a swivel is essential. We employed a rotary-assisted photometry system that has no light path interruptions (Doric Lenses). In addition, black acrylic powder was mixed in the dental cement to assure that ambient light was not leaking into the implant and reaching the optical fiber; this was tested in each animal by comparing fluorescence signals in the dark versus normal cage illumination.

## Optogenetics

The implanted optical fibers were connected to patch cables using sleeves. A black aluminum cap covered the head implant and completely blocked any light exiting at the ferrule's junction. Furthermore, the experiments occurred in a brightly lit cage that made it difficult to detect any light escaping the implant. The other end of the patch cable was connected to a dual light swivel (Doric lenses) that was coupled to a green laser (520 nm; 100 mW) to activate Arch or a blue laser (450 nm; 80 mW) to activate ChR2. Unless otherwise noted, the behavioral experiments employed green light between 25 and 35 mW and blue light between 1 and 3 mW. Power is regularly measured by flashing the connecting patch cords onto a light sensor – with the sleeve on the ferrule.

## Exploratory behavior and video tracking

During open-field experiments, mice are placed in a circular open field (10″ diameter) that was illuminated from the bottom or in a standard shuttle box (16.1″ × 6.5″). All mice in the study (open field or shuttle box) were continuously video tracked (30–100 FPS) in synchrony with the procedures and other measures. We automatically tracked head movements with two color markers attached to the head connector – one located over the nose and the other between the ears. The coordinates from these markers form a line (head midline) that serves to derive several instantaneous movement measures per frame (*Zhou et al., 2023*). Overall head movement was separated into *rotational* and *translational* components. Rotational movement was the angle formed by the head midline between succeeding video frames multiplied by the radius. Translational movement resulted from the sum of linear (forward vs backward) and sideways movements. *Linear* movement was the distance moved by the ears marker between succeeding frames multiplied by the cosine of the angle formed by the line between these succeeding ear points and the head midline. *Sideways* movement was calculated as linear movement, but the sine was used instead of the cosine. Pixel measures were converted to metric units using calibration and expressed as speed (cm/s). We used the time series to extract window measurements around events (e.g., CS presentations). Measurements were obtained from single trial traces and/or from traces averaged over a session. In addition, we obtained the direction of the rotational movement with a *Head Angle* or *bias* measure, which was the accumulated change in angle of the head per frame (vs the previous frame) zeroed by the frame preceding the stimulus onset or event (this is equivalent to the rotational speed movement in degrees). The *time to peak* is when the extrema occur versus event onset.

To detect spontaneous turns or movements from the head tracking, we applied a local maximum algorithm to the continuous head angle or speed measure, respectively. Every point is checked to determine if it is the maximum or minimum among the points in a range of 0.5 s before and after the point. A change in angle of this point >10° was a detected turn in the direction of the sign. We further sorted detected turns or movements based on the timing of previous detected events.

## Active avoidance tasks

Mice were trained in a signaled active avoidance task, as previously described (*Hormigo et al., 2016*; *Hormigo et al., 2019*). During an active avoidance session, mice are placed in a standard shuttle box (16.1″ × 6.5″) that has two compartments separated by a partition with side walls forming a doorway that the animal must traverse to shuttle between compartments. A speaker is placed on one side, but the sound fills the whole box and there is no difference in behavioral performance (signal detection and response) between sides. A trial consists of a 7-s avoidance interval followed by a 10-s escape interval. During the avoidance interval, an auditory CS (8 kHz, 85 dB) is presented for the duration of the interval or until the animal produces a conditioned response (avoidance response) by moving to the adjacent compartment, whichever occurs first. If the animal avoids by moving to the next

compartment, the CS ends, the escape interval is not presented, and the trial terminates. However, if the animal does not avoid, the escape interval ensues by presenting white noise and a mild scrambled electric foot-shock (0.3 mA) delivered through the grid floor of the occupied half of the shuttle box. This US readily drives the animal to move to the adjacent compartment (escape response), at which point the US terminates, and the escape interval and the trial ends. Thus, an *avoidance response* will eliminate the imminent presentation of a harmful stimulus. An *escape response* is driven by presentation of the harmful stimulus to eliminate the harm it causes. Successful avoidance warrants the absence of harm. Each trial is followed by an intertrial interval (duration is randomly distributed; 25–45 s range), during which the animal awaits the next trial. We employed four variations of the basic signaled active avoidance procedure termed AA1, AA2, AA3, and AA4.

In AA1, mice are free to cross between compartments during the intertrial interval; there is no consequence for ITCs.

In AA2, mice receive a 0.2-s foot-shock (0.3 mA) and white noise for each ITC. Therefore, in AA2, mice must passively avoid during the intertrial interval by inhibiting their tendency to shuttle between trials. Thus, during AA2, mice perform both signaled active avoidance during the signaled avoidance interval (like in AA1) and unsignaled passive avoidance during the unsignaled intertrial interval.

In AA3, mice are subjected to a CS discrimination procedure in which they must respond differently to a CS1 (8 kHz tone at 85 dB) and a CS2 (4 kHz tone at 70 dB) presented randomly (half of the trials are CS1). Mice perform the basic signaled active avoidance to CS1 (like in AA1 and AA2), but also perform signaled passive avoidance to CS2, and ITCs are not punished. In AA3, if mice shuttle during the CS2 avoidance interval (7 s), they receive a 0.5-s foot-shock (0.3 mA) with white noise and the trial ends. If animals do not shuttle during the CS2 avoidance interval, the CS2 trial terminates at the end of the avoidance interval (i.e., successful signaled passive avoidance).

In AA4, three different CSs, CS1 (8 kHz tone at 81 dB), CS2 (10 kHz tone at 82 dB), and CS3 (12 kHz tone at 82 dB) signal a different avoidance interval duration of 4, 7, and 15 s, respectively. Like in AA2, mice are punished for producing ITCs. In AA4, mice adjust their response latencies according to the duration of the avoidance interval signaled by each CS.

There are three main variables representing task performance. The percentage of active avoidance responses (% avoids) represents the trials in which the animal actively avoided the US in response to the CS. The response latency (latency) represents the time (s) at which the animal enters the safe compartment after the CS onset; avoidance latency is the response latency only for successful active avoidance trials (excluding escape trials). The number of crossings during the intertrial interval (ITCs) represents random shuttling due to locomotor activity in the AA1 and AA3 procedures, or failures to passively avoid in the AA2 procedure. The SPL of the auditory CS's were measured using a microphone (PCB Piezotronics 377C01) and amplifier (×100) connected to a custom LabVIEW application that samples the stimulus within the shuttle cage as the microphone rotates driven by an actuator controlled by the application.

## Spherical treadmill recordings

To record single-unit zona incerta activity during movement on a spherical treadmill (20 cm ball floating in air) mice underwent surgery (as described above) to implant a head-holder and create a craniotomy above zona incerta. The craniotomy was filled with silicone gel (Kwik-Sil), which was replaced during each recording session. During the surgery, several visible reference points were made on the implant to allow targeting the zona incerta using coordinates from the bregma-lambda plane after the surgery. Head-fixed mice were adapted to the spherical treadmill for a few days. The position of the animal in virtual space was monitored with two ball sensors that provide movement in the horizontal and vertical planes (*x*, *y* coordinates) using a custom LabVIEW (National Instruments) program that also controls stimulus delivery.

On the spherical treadmill, we delivered various stimuli including an auditory tone (8 kHz tone at ~85 dB), white noise, and air-puffs (50 ms pulses; 30 PSI) to the contralateral or ipsilateral whiskers and lower back. Mice moved on the treadmill spontaneously or motivated by trains of air-puffs (at 10 Hz) to the lower back the mouse. During recording sessions, we used the reference points taken during the surgery to lower a sharp glass pipette (2–6 MΩ, filled with saline) into the zona incerta to record from well-isolated single units while mice moved on the treadmill and/or sensory stimuli were applied. To maximize the chances of recording from zona incerta, we targeted its central portion in space with

some variation (zona incerta spans tip-to-tip about 2 × 2 × 0.3 mm *x*, *y*, *z*; the *z* depth is from its dorsal border at the central portion). Mice were recorded on multiple days in 2–3 hr sessions. Each unit was recorded for at least 60 trials of each stimulus. During the last recording session, a marking lesion was made at the top border of the zona incerta, which verified the accuracy of the coordinates used during the recording sessions. After correcting the coordinates based on the marking lesion location, the units were ascribed to the zona incerta if they were within the estimated zona incerta depth.

## Histology

Mice were deeply anesthetized with an overdose of ketamine. Upon losing all responsiveness to a strong tail pinch, the animals were decapitated, and the brains were rapidly extracted and placed in fixative. The brains were sectioned (100 µm sections) in the coronal or sagittal planes. Some sections were stained using Neurotrace. All sections were mounted on slides, cover slipped with DAPI mounting media, and all the sections were imaged using a slide scanner (Leica Thunder). We used an APP we developed with OriginLab (Brain Atlas Analyzer) to align the sections with the Allen Brain Atlas Common Coordinate Framework (CCF) v3 (*Wang et al., 2020b*). This reveals the location of probes and fluorophores versus the delimited atlas areas. For example, we used it to delimit Neurotrace-stained sections of control and dtA-lesion mice and count neurons within zona incerta.

## Acknowledgements

Supported by NIH grants to MAC. Additional information at https://castro-lab.org/.

## Additional information

### Funding

| Funder | Grant reference number | Author |
| --- | --- | --- |
| National Institute of Neurological Disorders and Stroke | R35 NS097272 | Manuel Castro-Alamancos |
| National Institute of Neurological Disorders and Stroke | R01 NS104810 | Manuel Castro-Alamancos |

The funders had no role in study design, data collection, and interpretation, or the decision to submit the work for publication.

### Author contributions

Sebastian Hormigo, Ji Zhou, Sarmad Sajid, Natan Busel, Formal analysis, Investigation, Writing – review and editing; Dorian Chabbert, Formal analysis, Investigation; Manuel Castro-Alamancos, Conceptualization, Data curation, Software, Formal analysis, Supervision, Funding acquisition, Investigation, Methodology, Writing - original draft, Project administration, Writing – review and editing

### Author ORCIDs

Natan Busel (iD) http://orcid.org/0000-0002-4525-0084
Manuel Castro-Alamancos (iD) http://orcid.org/0000-0002-2916-9585

Joint Public Review: https://doi.org/10.7554/eLife.89366.3.sa1
Author Response https://doi.org/10.7554/eLife.89366.3.sa2

## Additional files

### Supplementary files
• MDAR checklist

## Data availability

The data is shared on Dryad at https://doi.org/10.5061/dryad.18931zd42.

The following dataset was generated:

| Author(s) | Year | Dataset title | Dataset URL | Database and Identifier |
|---|---|---|---|---|
| Castro-Alamancos M, Hormigo S, Zhou J, Chabbert D, Sajid M, Busel N | 2023 | Zona incerta distributes a broad movement signal that modulates behavior | https://doi.org/10.5061/dryad.18931zd42 | Dryad, 10.5061/dryad.18931zd42 |

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
