## [Editor Report · eLife assessment]

This **important** study uses a range of technical approaches to investigate the responses of zona incerta neurons to movement and sensory stimuli. The majority of neurons exhibited movement related activity but only a small proportion were modulated by whisker deflections. The major conclusion of the study is that the zona incerta distributes a general motor signal. The evidence supporting this claim is **solid**, although the study would be improved by greater transparency and discussion of experimental methods and histological verification of recording sites, viral spread, and which territories of the zona incerta were investigated. The work will be of interest to behavioral and physiological neuroscientists.

---

## [Referee Report · Joint Public Review]

The manuscript presents compelling evidence for the role of the zona incerta area of the brain in regulating movement and sensory stimuli in mice. The study uses appropriate and validated methodology in line with the current state-of-the-art, including optogenetic manipulation and recording of single-unit activity. The authors' claims and conclusions are well-supported by their data, which includes a comprehensive review of previous research on the zona incerta. Overall, the manuscript provides solid evidence for the role of the zona incerta in regulating movement and sensory processing.

Major strengths and weaknesses of the methods and results.

The zona incerta have many integrative functions that link sensory stimuli with motor responses to guide behavior.

The study explored the activation of zona incerta GABAergic neurons during cued avoidance tasks and found that these neurons activate during goal-directed avoidance movement. Optogenetic manipulation of these neurons affected movement speed and performance during active avoidance tasks.

The findings suggest that the zona incerta area of the brain plays a significant role in regulating movement and responding to salient auditory tones in association with movement in mice. The evidence presented is fundamental and provides a comprehensive review of previous research on the zona incerta and its involvement in various behaviors and sensory processing.

The article is very well written, with a correct hypothesis and a cutting-edge methodology to achieve the expected objectives. Moreover, they use statistical rigorous approaches in the analysis of the results. Also, analyzes are performed using scripts that automate all aspects of data analysis, ensuring their objectivity. The results are very novel, and provides solid evidence for the role of the zona incerta in regulating movement and sensory processing.

---

## [Author Response]

The following is the authors’ response to the original reviews.

Thank you for reviewing and assessing our paper. Reviewer2 had only posive comments. Reviewer 1 also had posive comments but included a list of suggesons. The revised version includes text edits to address the suggesons.

**Reviewer 1:**
… First, it is unclear whether the experiments and analyses were set up to be able to rule out more specific candidate funcons of the ZI.

The list of possible funcons performed by the ZI is broad. Nevertheless, our study considers a rather long list of neural processes related to the behaviors listed below.

Second, many important details of the experiments and their results are hard to decipher given the current descripons and presentaons of the data.

The procedures used in the present study have all been used and described in our previous studies(cited). We used the same descripons and presentaons as in the prior studies. We have gone over the Methods and figures to ensure that all details required to understand the experiments are provided, but we also added further details following the suggesons noted below.

The paper could be significantly strengthened by including more details from each experiment, stronger jusficaons for the limited behaviors and experimental analyses performed, and, finally, a broader analysis of how the recorded acvity in the ZI relates to behavioral parameters.

The paper studied several behaviors including: (1) spontaneous movement of head-fixed mice on a spherical treadmill, (2) tacle (whisker, and body parts) and auditory (tones and white noise) smuli applied to head fixed mice, (3) spontaneous movement iniaon, change, and turns in freely moving mice, (4) auditory tone (frequency and SPL) mapping in freely behaving mice, (5) auditory-evoked orienng head movements (responses) in the context of several behavioral tasks, (6) signaled acve avoidance responses and escapes (AA1), (7) unsignaled/signaled passive avoidance responses (AA2ITI/AA3-CS2), (8) sensory discriminaon (AA3), (9) CS-US interval ming discriminaon (AA4), and (10) USevoked unsignaled escape responses.

In freely moving experiments, the behavior is connuously tracked and decomposed into translaonal and rotaonal movement components. Discrete responses are also evaluated (e.g., acve avoids, escapes, passive avoids, errors, intertrial crossings, latencies, etc.). These behavioral procedures evaluate many neural processes, including decision making (Go/NoGo in AA1-3), response control/inhibion (unsignaled and signaled passive avoidance in AA2/3), and smulus discriminaon (AA3). The applied smuli, discrete responses, and tracked movement are always related to the recorded ZI acvity using a variety of techniques (e.g., cross-correlaons, PSTHs, event-triggered me extracons, etc.), which relate the discrete and me-series parameters to the neural acvity. We do not think all this qualifies as, “limited behaviors”.

(1) Anatomical specificaon: The ZI contains many disnct subdivisions--each with its own topographically organized inputs/outputs and putave funcons. The current manuscript doesn't reference these known divisions or their behavioral disncons, and one cannot tell exactly which poron(s) of the ZI was included in the current study. Moreover, the elongated structure of the ZI makes it very difficult to specifically or completely infect virally. The data could be beter interpreted if the paper included basic informaon on the locaons of recordings, the extent of the AAV spread in the ZI in each viral experiment, and what fracon of infected neurons were inside versus outside ZI.

Our experiments employed Vgat-Cre mice to target ZI neurons. In this line, GABAergic neurons from the enre ZI express Cre, including the dorsal and ventral subdivisions (see (Vong et al., 2011; Hormigo et al., 2020)). Consequently, AAV injecons in Vgat-Cre mice produce restricted expression in the ZI that can fully delineate the nucleus as shown in the papers referenced above (including ours). There is nil expression in structures above or below ZI because they do not express Cre in these mice (e.g., thalamus and subthalamic nucleus), which allows for selecve targeng of ZI. Our optogenec manipulaons and photometry recordings were not aimed at specific ZI subdivisions. We targeted the area of ZI indicated by the stereotaxic coordinates (see Methods), which are aimed at the center of the structure to maximize success in recording/manipulang neurons within ZI. While all the animals included in the study expressed opsins and GCaMP within ZI that in many animals fully delineated the nucleus, there was normal variability in the locaon of opcal fibers, but we did not detect any differences in the results related to these variaons.

Fiber photometry and optogenecs experiments are performed with rather large diameter opcal probes, which record/manipulate relavely large areas of the targeted structure. This is useful because our goal was to idenfy funconal roles of the enre ZI, which could then be parsed. In the present study, we did not perform experiments to target specific ZI populaons (e.g., retrograde Cre expression from target areas), which may have revealed differences atributed to their projecon sites. However, in the last experiment, we selecvely excited ZI fibers targeng three different areas (midbrain tegmentum, superior colliculus, and posterior thalamus), which revealed clear differences on movement. Thus, future experiments should explore these different populaons (e.g., using retrograde/anterograde expression systems), which may be in different subdivisions.

We have enhanced the Methods secon to clarify these points, including the addion of these references.

(2) Electrophysiological recording on the treadmill: The authors are commended for this technically very difficult experiment. The authors do not specify, however, how they knew when they were recording in ZI rather than surrounding structures, parcularly given that recording site lesions were only performed during the last recording session. A map of the locaons of the different classes of units would be valuable data to relate to the literature.

We have added details about this procedure in the Methods secon. These recordings are performed based on coordinates, and categorizing neurons as belonging to ZI is obviously an esmate based on the final histological verificaon. Nevertheless, the marking lesions revealed that the electrodes were on target, which likely resulted from the care taken during the surgical procedure to define reference points used later during the recording sessions (see Methods). Regarding a map of the unit locaons, we performed several analyses that did not reveal clear differences based on site. For example, we compared depth vs cell class, “There was no difference in recording depth between the four classes of neurons (ANOVA F(3,337) = 1.06 p=0.3676)”. Future experiments that employ addional methods (labelling, opto-tagging, etc.) would be more appropriate to address mapping quesons. Finally, as we state in the paper, “However, these recordings do not target GABAergic neurons and may sample some neurons in the tissue surrounding the zona incerta. Therefore, we used calcium imaging fiber photometry to target GABAergic neurons in the zona incerta”.

(3) The raonale of the analysis of acvity with respect to “movement peak”: It is unclear why the authors did not assess how ZI acvity correlates with a broad set of movement parameters, but rather grouped heterogeneous behavioral epochs to analyze firing with respect to “movement peaks”.

The reviewer is referring to movement peaks on the spherical treadmill. On the treadmill, we used the forward locomotor movement of the animal because this is the main acvity of the mice on the treadmill. We considered “all peaks” (or movements) and “>4 sec peaks”, which select for movement onsets. Compared to the treadmill, in freely movement condions during various behavioral tasks, there is a richer behavioral repertoire, which was analyzed in more detail (i.e., translaonal, and rotaonal components during spontaneous ongoing movement and movement onsets, movement related to various behaviors such as orienng, acve and passive avoidance, escape, sensory smulaon, discriminaon, etc.). Thus, we focused on a broader set of movement parameters in the Cre-defined ZI cells of freely behaving mice.

(4) The display of mean categorical data in various figures is interesng, however, the reader cannot gather a very detailed view of ZI firing responses or potenal heterogeneity with so litle informaon about their distribuons.

The PCA performs the heterogeneity classificaon in an unbiased manner, which we feel is a thoughul approach. The firing rates and correlaons with movement for each category of neurons are detailed in the results. Furthermore, the sensory responses for these neurons are also detailed. Together, we think this provides a detailed view of the units we recorded in awake/head-fixed mice. As already stated, further study would benefit from an addional level of cell site verificaon.

(5) Somatosensory firing responses in ZI: It is unclear why the authors chose the specific smuli used in the study. How oen did they evoke reflexive motor responses? What was the latency of sensory-evoked responses in ZI acvity and the latency of the reflexive movement?

These are broad quesons, and we assume that the reviewer is asking about somatosensory evoked responses on the spherical treadmill. We used air-puffs applied to the whiskers and on the back (le vs right) because the whiskers represent an important sensory representaon for mice, and the back is a part of the body (trunk), which we oen use to movate the animals to move forward on the treadmill. Regarding the latency of the somatosensory evoked responses, in this case, we did not correct them based on the me it takes the air-puff to travel to the whiskers or body part, and therefore we did not provide latencies. Moreover, air-puffs are not a very good method to quanfy whisker-evoked latencies, which are beter measured using other methods (whisker deflecons of single/mulple whiskers using piezo-devices or other mechanical devices, as we and others have done in many studies). We are not sure what the reviewer means by “reflexive behavior”; we did not measure any reflexive behavior under these condions. We have gone over the Methods and Results to ensure that sufficient details are provided about these experiments.

(6) It would be valuable to see example traces in Figure 3 to get a beter sense of the me course and contexts under which Ca signals in ZI tracks movement. What is the typical latency? What is the typical range of magnitudes of responses? Does the Ca signal track both fast and slow movements? How are the authors sure that there are no movement arfacts contribung to the calcium imaging? It seems there is more informaon in the dataset that could be valuable.

As is well known, fiber photometry calcium imaging is a slow populaon signal. We do not think it would be valuable to get into ming issues beyond what is already detailed in the study (i.e., magnitudes measured as areas or peaks, and ming as me-to-peaks). Regarding “movement arfacts”, these signals are absent (flat) in animals that do not express GCAMP. We agree that there must be addional valuable informaon in our datasets (as in most me-series). However, the current paper is already rather extensive. We will connue to peruse our datasets and report addional findings in new papers.

(7) Figure 4: The raonale for quanfying the F/Fo responses over a 6-second window, rather than with respect to discrete movement parameters, is not well explained. What types of movement are binned in this approach and might this broad binning hinder the ability to detect more specific relaonships between acvity and movement?

Figure 4 is focused on characterizing the relaonship between turns (ipsiversive and contraversive) during movement and ZI acvity. We tested different binning windows to find differences, including the 6 sec window in figure 4 for populaon measures (-3 to 3 sec around the turns). This binning approach is effecve at revealing differences where they exist (e.g., superior colliculus) as shown in our previous studies (e.g. (Zhou et al., 2023)). Moreover, the turns in the different direcons can be considered discrete responses at their peak, and the ming of the related acvaons (e.g., me to peaks), which we evaluated, are rather sensive and would have revealed differences, but we did not find them.

(8) Separaon of sensory and motor responses in Figure 5: The current data do not adequately differenate whether the responses are sensory or motor given the high correlaon of the sensory inputs driving motor responses. Because isoflurane can diminish auditory responses early in the auditory pathway, this reviewer is not convinced the isoflurane experiments are interpretable.

The reviewer is referring to Fig. 5C,D. Indeed, the point of this experiment was to show that it is difficult to differenate whether neural responses are sensory or motor in awake and freely moving condions. As we stated in the Results secon, “Although arousal and movement were not dissected in the present experiment (this would likely require paralyzing and ventilating the animal), the results indicate that activation of zona incerta neurons by sensory stimulation is primarily associated with states when sensory-evoked movement is also present”. This is followed in the Discussion by, “…as already noted, the suppression of sensory responses may be due to changes in arousal (Castro-Alamancos, 2004; Lee and Dan, 2012) and not caused by the abolishment of the movements per se”.

(9) Given the broad duraon of the mean avoidance response (Fig. 6 C, botom), it would be useful to know to what extent this plot reflects a prolonged behavior or is the result of averaging different animals/trials with different latencies. Given that the shapes of the F/Fo responses in ZI appear similar across avoids and escapes (Fig. 6D), despite their apparent different speeds and movement duraons (Fig 6C), it would be valuable to know how the ming of the F/Fo relates to movement on a trial-by-trial basis.

The duraon of the avoidance response cannot be ascertained from CS onset (panel 6C botom) and avoids are not wide but rather sharp. We have now made this clearer when Fig. 6C is first menoned (“note that since avoids occur at different latencies after CS onset they are best measured from their occurrence as in Fig. 6D”). Like other related condioned and uncondioned responses, avoids and escapes are similar, varying in the noted parameters. Regarding ming, as already menoned above, we think that the characteriscs of the populaon calcium signal make it unsuitable for further ming consideraons than what we included, parcularly for movements occurring at the fast speeds of avoids and escapes.

(10) Lesion quanficaon: One cannot tell what rostral-caudal extent of ZI was lesioned and quanfied in this experiment. It would be easier to interpret if also ploted for each animal, so the reader can tell how reliable the method is. The mean ablaon would be beter shown as a normalized fracon of cells. Although the authors claim the lesions have litle impact on behavior, it appears the incompleteness of the lesions could warrant a more conservave interpretaon.

The lesion experiment was a complement to the optogenecs inacvaon experiments we performed in our preceding ZI paper and in the present paper. Thus, the finding that the lesions had litle impact on behavior is supporve of the optogenecs findings. Regarding cell counts, we did not select any parts of the ZI to quanfy the number of neurons in either control or lesion mice. We considered the full rostrocaudal extent in our measurements. We are not sure what “fracon” the reviewer is suggesng, considering that these counts are from two different groups of mice (control vs lesion). Note that the red-marked neurons, as shown in Fig. 8A, reveal healthy non-Vgat-Cre neurons outside ZI that mark the extent of the AAV diffusion, which as shown spanned the full extent of the ZI in the coronal plane (and in other planes as the AAV spreads in all direcons).

(11) Optogenecs: the locaon of infected neurons is poorly described, including the rostral-caudal extent and the fracon of neurons inside and outside of ZI. Moreover, it is unclear how strongly the optogenec manipulaons in this study are expected to affect neuronal acvity in ZI.

We discussed the first point in (1) above. Regarding, how optogenec manipulaons are expected to affect neuronal acvity in ZI and its targets, we have conducted extensive electrophysiological recordings in slices and in vivo to detail the effects of our manipulaons on GABAergic neurons (e.g. Hormigo et al., 2016; Hormigo et al., 2019; Hormigo et al., 2021a; Hormigo et al., 2021b), including ZI neurons (Hormigo et al., 2020). In fact, we never use an opsin we have not validated ourselves using electrophysiology. Moreover, our experiments employ a spectrum of optogenec light paterns (including trains/cont at different powers) that trate the optogenec effects within each session/animal. As shown in fig. 11 and 12, these paterns produce different behavioral effects related to the different levels of neural firing they induce. For ChR2-expressing neurons in ZI, firing is frequency dependent and maximal during Cont blue light (at the same power). For Arch-expressing neurons only Cont is used, and inhibion is a funcon of the green light power. When blue light is applied in ZI fibers targeng different areas, this relaonship changes. Blue light trains (1-ms pulses) at 40-66 Hz become the most effecve means of inducing sustained postsynapc inhibion compared to Cont or low frequencies.

References

Castro-Alamancos MA (2004) Dynamics of sensory thalamocorcal synapc networks during informaon processing states. Progress in Neurobiology 74:213-247.

Hormigo S, Vega-Flores G, Castro-Alamancos MA (2016) Basal Ganglia Output Controls Acve Avoidance Behavior. J Neurosci 36:10274-10284.

Hormigo S, Zhou J, Castro-Alamancos MA (2020) Zona Incerta GABAergic Output Controls a Signaled Locomotor Acon in the Midbrain Tegmentum. eNeuro 7.

Hormigo S, Zhou J, Castro-Alamancos MA (2021a) Bidireconal control of orienng behavior by the substana nigra pars reculata: disnct significance of head and whisker movements. eNeuro. Hormigo S, Vega-Flores G, Rovira V, Castro-Alamancos MA (2019) Circuits That Mediate Expression of Signaled Acve Avoidance Converge in the Pedunculoponne Tegmentum. J Neurosci 39:45764594.

Hormigo S, Zhou J, Chabbert D, Shanmugasundaram B, Castro-Alamancos MA (2021b) Basal Ganglia Output Has a Permissive Non-Driving Role in a Signaled Locomotor Acon Mediated by the Midbrain. J Neurosci 41:1529-1552.

Lee SH, Dan Y (2012) Neuromodulaon of brain states. Neuron 76:209-222.

Vong L, Ye C, Yang Z, Choi B, Chua S, Jr., Lowell BB (2011) Lepn acon on GABAergic neurons prevents obesity and reduces inhibitory tone to POMC neurons. Neuron 71:142-154.

Zhou J, Hormigo S, Busel N, Castro-Alamancos MA (2023) The Orienng Reflex Reveals Behavioral States Set by Demanding Contexts: Role of the Superior Colliculus. J Neurosci 43:1778-1796.